# The role of non-axisymmetry of magnetic flux rope in constraining solar eruptions

Ze Zhong [1,2], Yang Guo [1,2 ✉] & M. D. Ding [1,2 ✉]

Whether a solar eruption is successful or failed depends on the competition between different components of the Lorentz force exerting on the flux rope that drives the eruption. The present models only consider the strapping force generated by the background magnetic field perpendicular to the flux rope and the tension force generated by the field along the flux rope. Using the observed magnetic field on the photosphere as a time-matching bottom boundary, we perform a data-driven magnetohydrodynamic simulation for the 30 January 2015 confined eruption and successfully reproduce the observed solar flare without a coronal mass ejection. Here we show a Lorentz force component, resulting from the radial magnetic field or the non-axisymmetry of the flux rope, which can essentially constrain the eruption. Our finding contributes to the solar eruption model and presents the necessity of considering the topological structure of a flux rope when studying its eruption behaviour.

[1] School of Astronomy and Space Science, Nanjing University, Nanjing, P.R. China. [2] Key Laboratory for Modern Astronomy and Astrophysics (Nanjing University), Ministry of Education, Nanjing, P.R. China. ✉email: guoyang@nju.edu.cn; dmd@nju.edu.cn

Magnetic flux ropes (MFRs) are a bundle of twisted field lines with electric currents flowing inside. They play a critical role in explaining a variety of phenomena in astrophysics[1], solar physics[2–5], space physics[6], and laboratory plasmas[7]. In the solar atmosphere, MFRs are regarded as one of the basic magnetic structures to host prominences[8,9], sigmoidal structures[10,11], coronal cavities[12,13], and hot channels[14,15]. MFRs can erupt under some conditions, which can in turn drive prominence eruptions and coronal mass ejections (CMEs)[16], as well as solar flares when magnetic reconnection occurs[17]. The propagation of CMEs into the interplanetary space can disturb the space environment and is a potential hazard to the high technology facilities of the modern society if they direct at the Earth. Therefore, study on the physical mechanisms of MFR eruptions are one of the central topics in solar and space physics.

MFR eruptions are usually thought to be driven by ideal magnetohydrodynamic (MHD) instabilities. For a single isolated MFR, both the helical kink instability and torus instability could drive its eruption, probably at different stages. The helical kink instability occurs when the safety factor, $q$, is lower than a critical value, $q_c$, or when the twist number measured in turns, $\mathcal{T}$, is greater than a threshold, $\mathcal{T}_c$, where $q = 1/\mathcal{T}$ and $q_c = 1/\mathcal{T}_c$ (see Methods subsection Magnetic field diagnosis). The classical Kruskal–Shafranov critical value for helical kink instability is $q_c = 1$ or $\mathcal{T}_c = 1$[18]. In fact, the critical values depend on the details of the MFR equilibrium, such as the radial profile of the magnetic field, plasma $\beta$, and the flux rope aspect ratio, $\epsilon = R_0/a$, where $R_0$ and $a$ are the major and minor radii of the MFR, respectively. The critical values vary in a range of $q_c \in [0.6, 1.0]$ or $\mathcal{T}_c \in [1.0, 1.7]$ depending on the parameter settings of the MFR[19,20]. On the other hand, the torus instability occurs when the external poloidal magnetic field decays fast enough along the eruption path of the MFR, namely, the decay index, $n$, is larger than a critical value, $n_c$ (see Methods subsection Magnetic field diagnosis). Assuming that the MFR current path is semicircular and without any internal structures, the stability analysis yields $n_c = 1.5$[21]. The critical value for the torus instability also depends on the details of the MFR equilibrium, including the shape of the current path. A range of $n_c \in [0.8, 1.5]$ is derived with different prescriptions of the MFR structure[22–25].

The MFR eruptions can be confined in the lower corona or they fail to propagate out of the solar corona. A well known mechanism is the failed kink instability. Namely, the twist of the erupting flux rope is high enough to incur helical kink instability, but the decay index of the background field does not exceed the critical value for torus instability along the erupting path[26]. Is there a case where the decay index exceeds the critical value, but the eruption of a flux rope is still confined? Observations have reported quite a number of such examples[27,28]. Recently, a theoretical explanation has been presented in a series of papers[29,30]. The authors pointed out that the previous torus instability model misses a key component of the Lorentz force, namely, the magnetic tension force generated by the toroidal magnetic field (also known as the guide field) and the poloidal electric current in the MFR. The magnetic tension force could direct downward, similarly to the strapping force from the external poloidal field perpendicular to the MFR axis. The combination of these two downwardly directed forces counterparts the upwardly directed hoop force of the MFR. Such a theory was termed as the failed torus regime[29], in which a dynamic magnetic tension force might confine an MFR eruption. A typical signature for the failed torus instability to work is that the poloidal magnetic field is rapidly converted to the toroidal field when the MFR erupts, therefore decreasing the twist number and increasing the safety factor. We note that, all the previously proposed models assume a certain degree of axisymmetry of the MFR in which only the poloidal and

toroidal components of the magnetic field are considered while the radial component is neglected. In real observations, the magnetic topology is very complex and the MFR is usually non-axisymmetric, so that all the components should be considered when evaluating the Lorentz force acting on the MFR.

In order to clarify the causes of failed torus instability, MHD simulations are needed to be performed to reproduce observations. However, there are a number of difficulties in this problem. First, we deal with a fully three-dimensional (3D), dynamically evolving plasma system, which rules out the stationary extrapolation techniques that are usually used. Second, it is very difficult to extract the key parameters responsible for the failed torus instability with pure observations, and thus the parameter space is too large to explore if parameterized simulations are used to fit the observations. Considering these, a state-of-the-art technique, namely, a data-driven or data-constrained MHD simulation, should be adopted to explore an observed failed eruption. Even so, there are still some technical difficulties in computing the key parameters, such as how to identify the boundary of an MFR (see Methods subsection Identification of the MFR) and how to compute the decay index and twist number more accurately.

Here we apply a data-driven MHD model[31] in the framework of Message Passing Interface Adaptive Mesh Refinement Versatile Advection Code (MPI-AMRVAC 2.0[32]) to a confined MFR eruption[33] and diagnose in detail the physical causes leading to the failure of the eruption with some elaborately designed diagnostic tools[34]. We show a Lorentz force component, induced by the non-axisymmetry (or the radial magnetic field component) of the MFR, which plays a major role in preventing the MFR from erupting successfully.

## Results

**Overview of the event.** The eruption event under study was hosted in NOAA Active Region (AR) 12268, which appeared on the visible side of the Sun on 21 January 2015. The AR was a bipolar cluster on the east solar limb. With the emergence and cancellation of the photospheric magnetic field, a parasitic positive polarity gradually emerged near the center of the negative polarity in the AR for several days. Subsequently, the magnetic configuration of the AR became complex. It produced six Geostationary Operational Environmental Satellite (GOES) M- and C-class flares[33] within 36 h from January 29 to 30. However, it is puzzling that none of the six flares produced any CME although the 3 M-class flares were very intense and erupted in a short interval. Here, we focus on the intense M2.0 flare (Fig. 1) that occurred at 00:32 UT on 30 January 2015, which was observed by the Atmospheric Imaging Assembly[35] (AIA) on board Solar Dynamics Observatory[36] (SDO). The rationale for our choice is that a special magnetic structure, namely, a bifurcated MFR, was formed before the eruption onset[33].

Figure 2a shows the AIA 171 Å extreme-ultraviolet image at 00:00 UT, overlaid with the full view of the reconstructed coronal magnetic fields before the flare onset. The reconstructed fields are extrapolated using a nonlinear force-free field model[37], based on the photospheric vector magnetic field taken 32 min before the flare onset by the Helioseismic and Magnetic Imager[38] (HMI) on board SDO. The extrapolated fields reach a stable state, which is as close as possible to an equilibrium state, and satisfy the force-free and divergence-free conditions very well. The magnetic configuration of the AR is appealing because it is complicated enough. As shown in Fig. 2b, the AR consists of three main polarities. A positive polarity (P1) in the center is partially enclosed by a negative one (N1 extending to N1e). Another remote positive polarity (P2) also connects to N1 through a bifurcated MFR. We mark the bifurcated and unbranched parts of

the MFR with yellow and orange field lines, respectively. There also exists a dome-like structure in the corona marked by cyan field lines. Although the bifurcated MFR has three footpoints, the strong current channel is mainly concentrated in the central unbranched part of the MFR. This current channel has two footpoints rooted in the photosphere. We also make the scatter plots of the twist number of all sample magnetic field lines against the distance from the MFR axis (Supplementary Fig. 1a and b). The maximum value of the absolute twist number of the MFR is larger than 1.6 turns, which is close to the maximum threshold for helical kink instability.

Observations at SDO/AIA 94, 171, and 1600 Å passbands (Fig. 3) show a clear process of the confined eruption. In particular, we present the running difference between two consecutive AIA 171 Å images in order to reveal more clearly the eruption structure. The emission at AIA 171 Å originates from the upper transition region and the quiet corona, which unveils the plasma with a characteristic temperature of $6.3 \times 10^5$ K. When the M2.0 flare occurred, a clear plasmoid-like structure ejected (Fig. 3a), and it was subsequently flowed to the polarity P2 as shown in the 171 Å images. We make four slices to track the motion of the plasma. As shown in Fig. 3b, the time-distance diagrams for slices 1 and 2 display plasma flows along the coronal loops towards polarity P2 near the flare peak time, while those for slices 3 and 4 perpendicular to the coronal loop show that the plasma rises to a certain height and then stops rising, indicating a failed eruption. Compared to emission at 171 Å, emission at 94 Å has a higher characteristic temperature of $6.3 \times 10^6$ K, which reveals a large current channel in the corona. The composite snapshot in Fig. 3c clearly displays that the emission at 94 Å is stronger than that at 171 Å in the region of the bifurcated MFR. One can see that the two contours delineating the strongest 94 Å emission coincide with the two footpoints of the current channel integrated along the vertical direction deduced from the extrapolated magnetic field. It is noted that the flare emission at 1600 Å shows a complex structure with six ribbons (Fig. 3d). Ribbons R1, R2, and R5 are semicircular in shape that enclose the polarity P1. Ribbons R2 and R4, positioned on both sides of the polarity inversion line (PIL), exhibit a clear shear and separate from each other with flare development, as predicted in the standard flare model. The intensity of ribbon R3 is weak, which appears later than the aforementioned ribbons. The emission characteristics of ribbon R6 at polarity P2 are similar to that of ribbons R2 and R4, suggesting that these ribbons are heated through the same energy release mechanism.

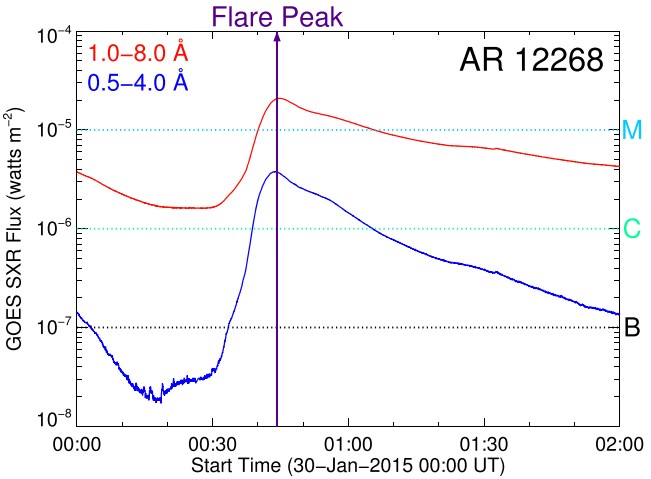

**Fig. 1 GOES soft X-ray fluxes of the M2.0 class solar flare on 30 January 2015 at 1.0–8.0 Å (red curve) and 0.5–4.0 Å (blue curve).** The purple vertical line shows the peak time of the flare at 00:44 UT. The three letters of M, C, and B represent the flare classes, according to the X-ray peak flux at 1.0–8.0 Å.

**3D MHD simulations.** To clarify the mechanism for the failed eruption, we perform a data-driven MHD simulation using the zero-$\beta$ approximation. The initial condition is reconstructed from a nonlinear force-free field[37] (see Methods subsection Initial and boundary conditions) based on the vector magnetic field observed by SDO/HMI. The boundary condition on the bottom is provided by the timeseries of vector magnetic fields and vector velocities (see Methods subsection Initial and boundary conditions).

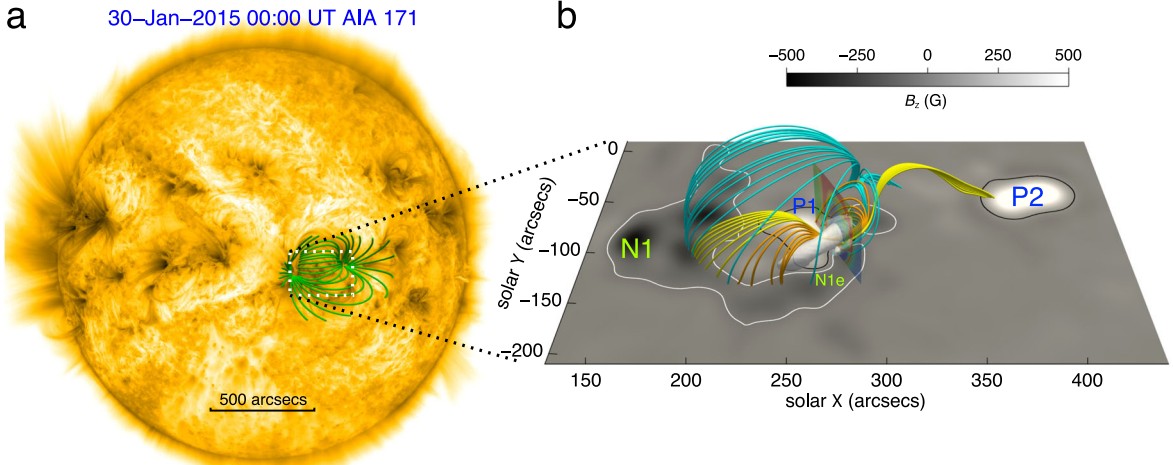

**Fig. 2 Overview of the M2.0 class solar flare observed on 30 January 2015. a** Full-disk image of AIA 171 Å at 00:00 UT on January 30. Some selected field lines (in green) at a relatively high altitude are overlaid in the active region. **b** The initial reconstructed magnetic field with a zoomed-in view of the region corresponding to the white dotted box in panel **a**. Yellow and orange lines represent the bifurcated and unbranched parts of the MFR, respectively. Cyan lines represent the dome-like structure. The white semi-transparent isosurface represents the electric current density that is larger than 32.9% of the maximum value in the whole domain. The transparent slice perpendicular to the MFR axis shows a cross section of the MFR delineated by the QSLs. The background image shows the magnetogram with polarities labeled as N1, N1e, P1, and P2, overlaid by the white and black contours with contour levels of $B_z$ ($B_z \equiv \mathbf{e}_z \cdot \mathbf{B}$) being $-50$ G and 50 G, respectively.

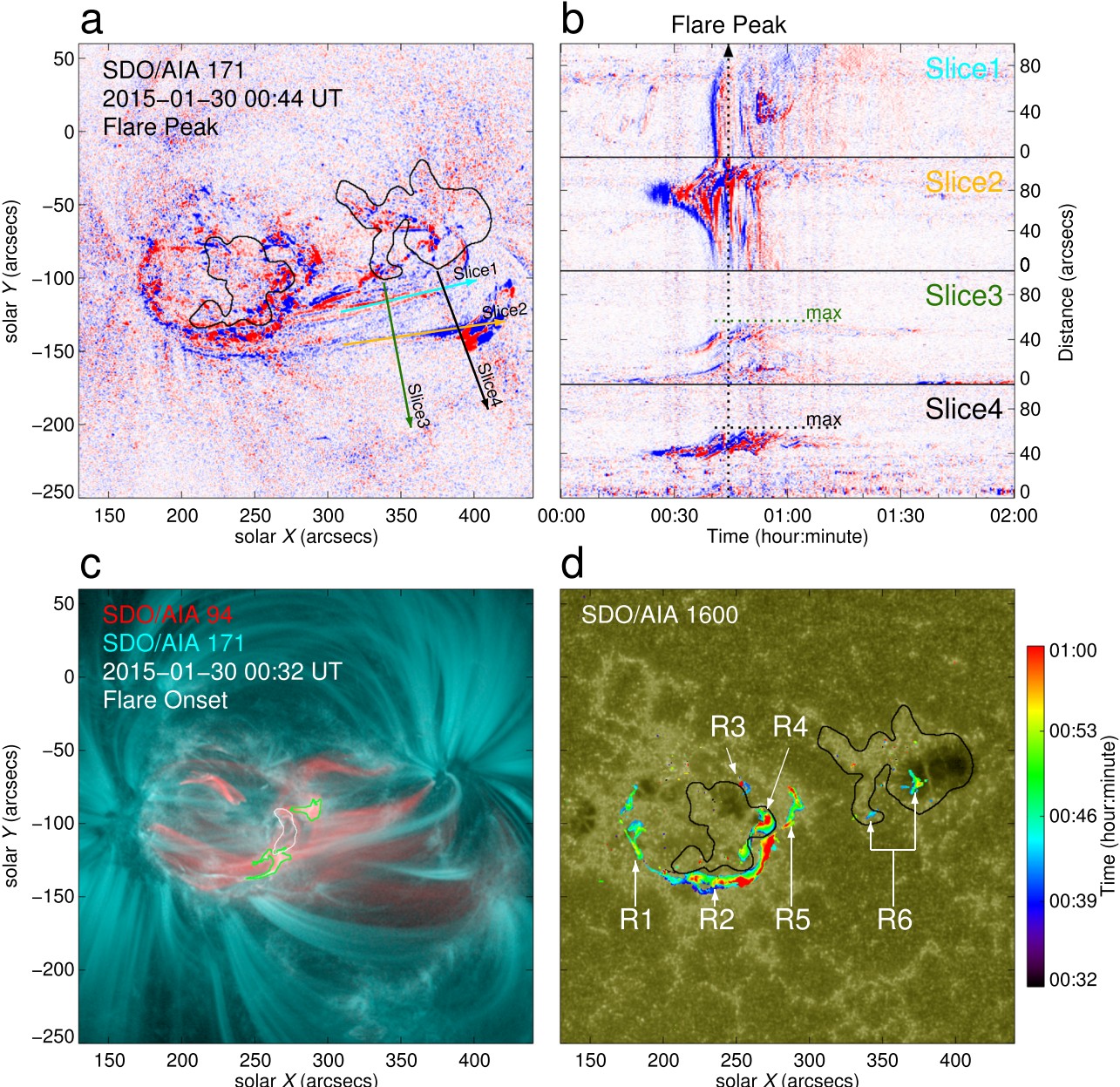

**Fig. 3 Flare emission in AIA multiwavelength channels. a** Running-difference image of AIA 171 Å showing the eruption structure at the peak time of the M2.0 flare, 00:44 UT, on 30 January 2015. The red and blue colors represent positive and negative differences in intensity, respectively. The black contour in the background shows the positive polarity with $B_z$ being +50 G. The cyan and orange lines show two different slices parallel to the coronal loop, while the green and black lines refer to the slices perpendicular to the coronal loop. **b** Time-distance diagrams showing the motion of the plasma along the four slices either parallel or perpendicular to the coronal loop. The black dotted line shows the flare peak time at 00:44 UT. **c** Composite image of AIA 94 Å (red) and 171 Å (cyan) at the onset time of the M2.0 flare, 00:32 UT, on 30 January 2015. The green and white contours represent 60% of the maximum emission at 94 Å and 50% of the maximum electric current integrated along the line of sight, respectively. **d** Timeseries of AIA 1600 Å emission starting from 00:32 UT. The black contour is the same as that in panel **a**. The arrows point to the flare ribbons, which are labeled as R1–R6.

Figure 4a–d exhibits the 3D dynamic evolution of the magnetic field and the electric current density at four typical moments covering the main period of the event. First, the bifurcated part of the MFR disappears when it rises initially. Subsequently, the unbranched part of the MFR continues to rise and further reconnects with the sheared field lines that surround it. During this process, several field lines below the MFR become a part of the MFR by magnetic reconnection. Their footpoints at one end, originally located in the parasitic polarity P1, are gradually transferred to the remote polarity P2. Meanwhile, the dome-like structure above is also stretched by the rising MFR. Although the whole MFR then expands to a very large size, the spatial distributions of four quantities, including the electric current, quasi-separatrix layer, twist number, and magnetic flux, indicate that the main body of the MFR is still complete and keeps its coherence. It is noted that a current sheet is stretched out under the MFR, which has been predicted in the standard solar flare model. Detailed dynamics of the MFR are shown in Supplementary Movie 1. We also measure the 3D height of the MFR front and its velocity from the MHD simulation. As shown in Fig. 5a, the MFR goes up at an almost constant velocity after its initial acceleration, and it then decelerates gradually after the flare

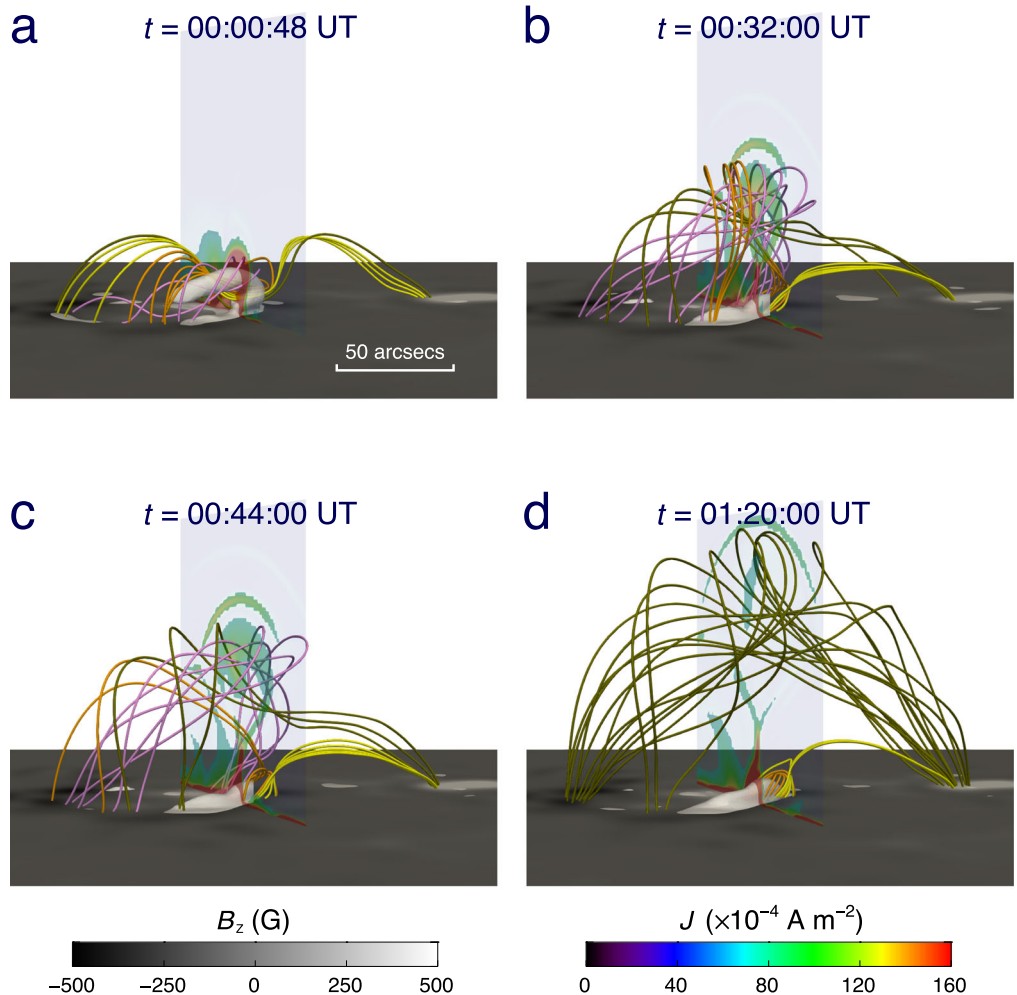

**Fig. 4 Snapshots showing the MFR and the electric current density. a–d** Four snapshots at 00:00, 00:32, 00:44, and 01:20 UT on 30 January 2015 corresponding to the pre-flare, flare onset, flare peak and end times, respectively. The yellow and orange lines have the same meaning as that in Fig. 2b. The pink and olive lines within the MFR represent the field lines connecting the negative polarity N1 and the positive polarities P1 and P2, respectively. The vertical transparent slice displays the electric current density. The white semi-transparent isosurface represents the electric current density larger than 32.9% of the maximum value in the whole domain. The background shows the distribution of the vertical magnetic field component, $B_z$.

peak, mimicking what has actually happened. We further study the temporal evolution of the toroidal flux of the MFR calculated by $\phi_T = \int \mathbf{B}_T \cdot d\mathbf{S}$, where $\mathbf{B}_T$ is the total toroidal magnetic field and $\mathbf{S}$ is the cross-sectional area of the MFR. We then get the poloidal flux by $\phi_P = \mathcal{T}\phi_T$ (see Methods subsection Magnetic field diagnosis for computation of the twist number $\mathcal{T}$). We find that during the eruption of the MFR, the toroidal flux increases continuously, while the poloidal flux first increases, then decreases after the flare peak time, and keeps almost unchanged in the late phase (Fig. 5b).

We then characterize the magnetic topology through comparing the simulation results with the AIA extreme-ultraviolet observations. First, we compare the 3D magnetic field with the AIA emission at 304 Å (Fig. 6a). We delineate typical field lines with different colors, which represent the MFR and the overlying field. The footpoints of field lines coincide well with the brightening ribbons of the flare. Second, we calculate the electric current density and identify some important magnetic topological structures, such as separators[39], hyperbolic flux tube[40] (HFT), and, more commonly, quasi-separatrix layers[41] (QSLs). QSLs are defined as places with strong gradients of magnetic connectivity. We usually identify the QSLs through a topological parameter called the squashing factor $Q$. Figure 6b shows a side view of the

MFR, which clearly displays the positions of high electric current density and that of QSLs ($\log(Q) > 3$). It is seen that a part of high electric current densities coincide well in space with high $Q$ values. Figure 6c shows the extreme-ultraviolet emission at 94 Å at the flare peak time, which indicates a hot channel structure along the PIL. We also make a synthetic map of the AIA emissivity by integrating the electric current density along the vertical direction at the flare peak time (Fig. 6d). It is seen that the emission along the PIL is more prominent at the position of the MFR.

It is known that QSLs (Fig. 7a, d) are also favorable locations where energetic particles can be accelerated and move downward along the field lines to heat the lower solar atmosphere, resulting in the brightening of flare ribbons. Previous studies have shown the spatial correspondence between flare ribbons and photospheric QSLs by using extrapolation models[42], magneto-frictional models[43,44], or MHD simulations[45,46]. However, when a flare occurs, the magnetic topology in the core of the flare region changes rapidly due to magnetic reconnection. Thus, the magnetic topology itself is very dynamic in nature. In practice, it is difficult to trace precisely the simulated QSLs following the observed flare ribbons due to an unavoidable gap between simulations and observations, such as some unpredictable

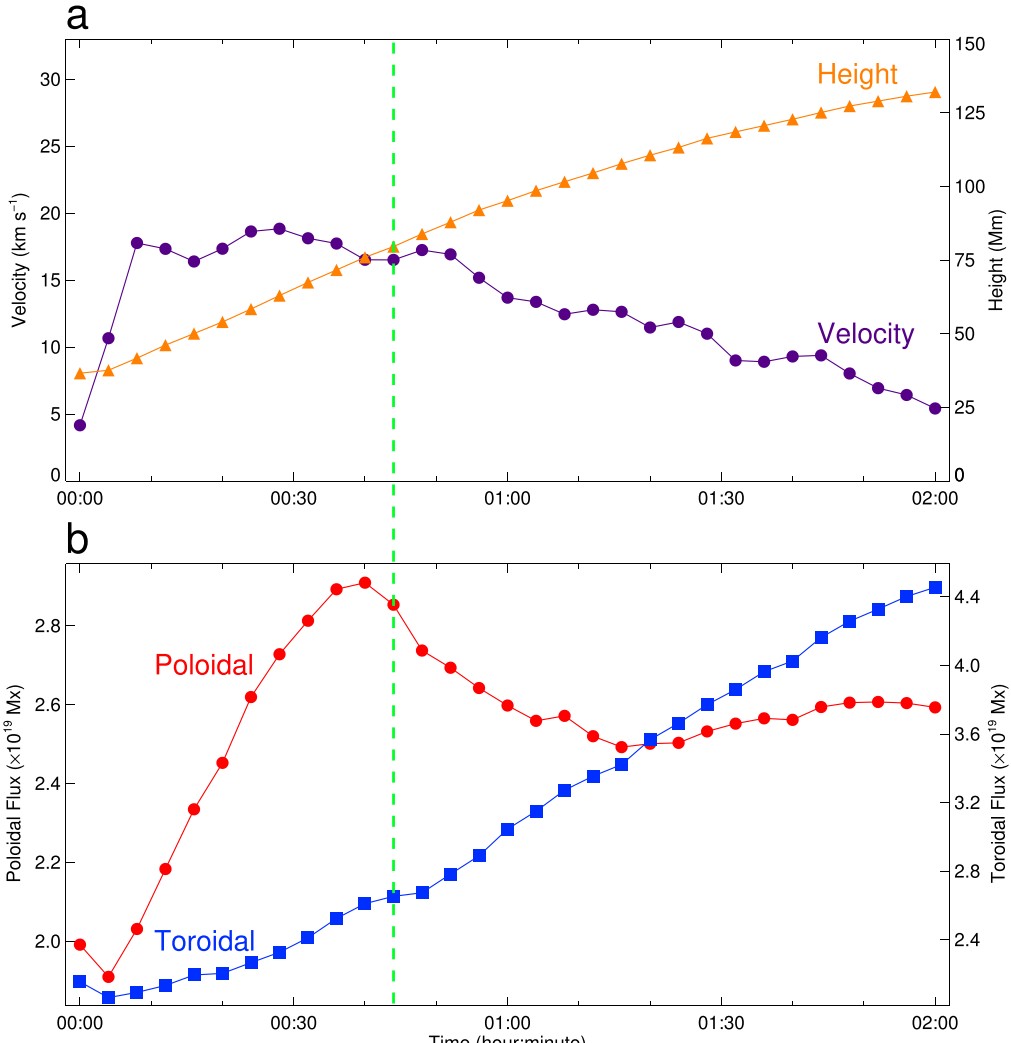

**Fig. 5 Kinematics and magnetic fluxes of the MFR. a** Time–height measurement of the MFR in the MHD simulation. The orange triangles and purple circles show the height and velocity, respectively. **b** Temporal evolutions of the toroidal flux (blue squares) and poloidal flux (red circles) of the MFR. The green dashed line represents the peak time of the M2.0 flare at 00:44 UT.

resistivity in the real corona but spurious numerical resistivity in simulations[47]. In spite of these difficulties, we try to find the dynamic relationship between the QSLs and flare ribbons. To this end, we compute the squashing factor $Q$ on the bottom and identify the signed QSLs with $|\log(Q)|>3$, which are then overlaid on the flare ribbons (Supplementary Figs. 2 and 3). Four typical moments, corresponding to the pre-flare, flare onset, flare peak and end times, respectively, are shown in order to reveal the dynamic evolution. We find that the evolution of the QSLs matches the flare ribbons very well. This means that our simulation reaches a high degree of realism at least in the dynamic evolution of the magnetic topology. Further quantitative analysis shows that the separation of the QSLs undergoes two stages. The separation velocity before the flare peak time is larger than that after the peak time, a dynamic behaviour which is almost the same as what is revealed by the observed flare ribbons. More details on the dynamics of the flare ribbons compared with the QSLs are shown in Supplementary Movie 2.

**Mechanism of the confined eruption.** Our model successfully reproduces the process of the failed eruption and can thus reveal which physical mechanisms finally halt the eruption. In a low $\beta$ plasma, where the gravity and thermal pressure can be omitted,

whether the MFR eruption is successful or failed depends on the competition between various Lorentz forces exerting on it, including the hoop force from the gradient of the magnetic pressure, the tension force from the toroidal magnetic field, the strapping force from the external poloidal magnetic field and the force caused by the radial magnetic field of the MFR. If the net Lorentz force is downward, the MFR cannot break through the overlying magnetic field to form a CME. To explore this problem quantitatively, we calculate the Lorentz force along the vertical direction, $L_z = \mathbf{e}_z \cdot (\mathbf{J} \times \mathbf{B})$. Figure 7b, e shows the spatial distribution of $L_z$ on a vertical cut of the simulation domain at 00:16 and 02:00 UT, respectively. We find that the direction of $L_z$ within the radius of the MFR is initially upward, facilitating the eruption, and it later becomes downward, thus preventing the MFR from further moving up. The overall distribution of $L_z$ at different positions in the MFR is downward after 00:52 UT. We also notice that the Lorentz force is still upward but the MFR has already a tiny deceleration around 00:40 UT. A possible reason is that the Lorentz force acts directly on the local plasma, while the apparent velocity of the MFR is related not only to the plasma velocity but also to the change of magnetic structure of the MFR, the latter of which is changing significantly during this period. The detailed temporal evolution of $L_z$ is shown in Supplementary Movie 3.

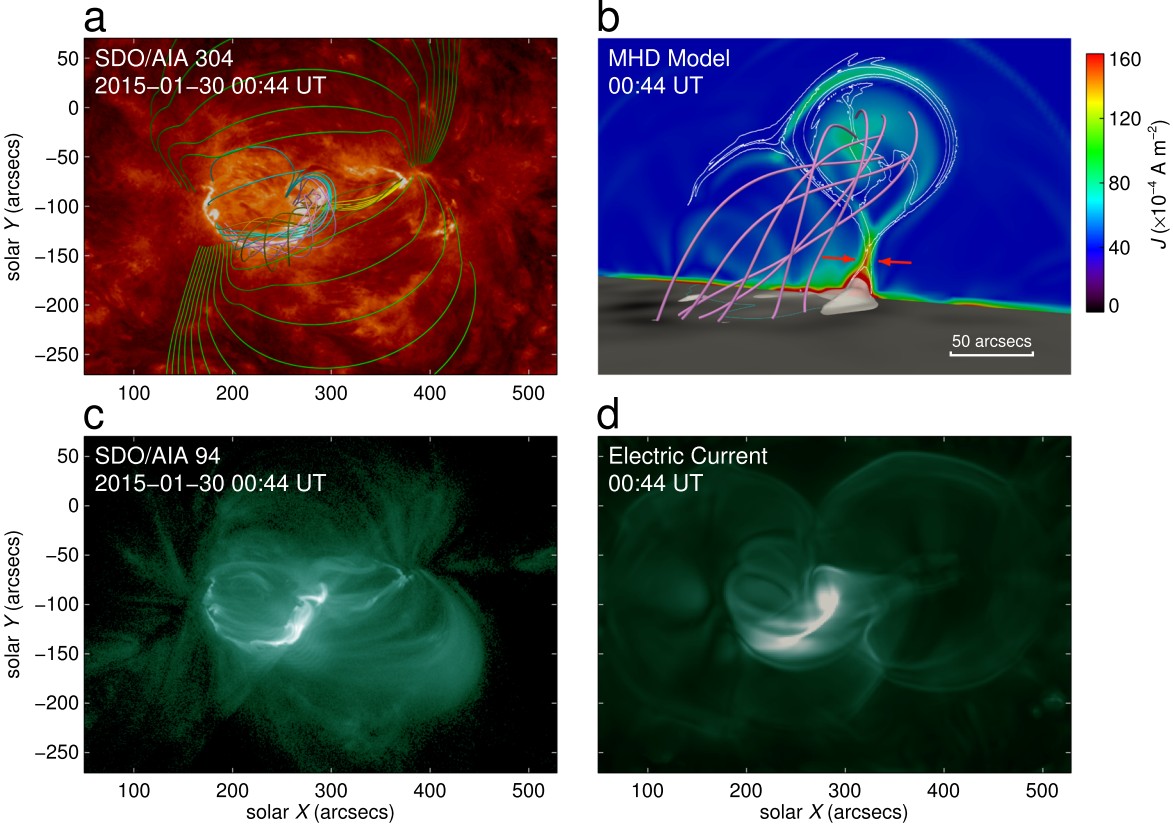

**Fig. 6 Comparison between MHD simulations and AIA extreme-ultraviolet observations at the peak time of the M2.0 flare. a** Selected magnetic field lines overlaid on an AIA 304 Å image from the vertical perspective. The field lines in green are located at a relatively high altitude. The pink and olive lines have the same meaning as that in Fig. 4. The cyan, yellow and orange lines have the same meaning as that in Fig. 2b. **b** The MFR from a side view. The field lines in pink are the same as that in panel **a**. The vertical slice perpendicular to the MFR axis displays the electric current density overlaid by the contour of the QSLs with log($Q$)>3. The two red arrows indicate the inflow at the reconnection site. The white semi-transparent isosurface represents the electric current density larger than 32.9% of the maximum value in the whole domain. The bottom surface shows the distribution of the vertical magnetic field component, $B_z$, overlaid by the polarity inversion line in cyan. **c** AIA 94 Å image at 00:44 UT on 30 January 2015. **d** Synthetic emission at 94 Å calculated by the integration of the electric current density along the vertical direction at the peak time.

We then decompose the total vertical Lorentz force $L_z$ into different components contributed by various combinations of the magnetic field and electric current components, as listed in Supplementary Table 1 and Supplementary Fig. 4a and b. Figure 7c, f shows the distribution of the different force components along a vertical line perpendicular to the MFR axis at 00:16 and 02:00 UT, respectively. As in previous models, the hoop force ($F_H$) always directs upward below the MFR axis, acting as a driving force of the eruption. More interestingly, we find that at the time of 02:00 UT, three force components direct downward below the MFR axis, including the magnetic tension force ($F_T$), and the forces induced by the radial magnetic field component of the MFR ($F_{N1}$ and $F_{N2}$). The former force does not change its direction as shown in Supplementary Fig. 4c. The latter forces are not included in previous models. Note that the radial field component deforms the cross section and thus leads to a non-axisymmetry of the MFR. We thus call these two forces the non-axisymmetry induced forces. In particular, $F_{N1}$ changes its direction during the eruption process. It is initially upward, pushing the MFR rising, but it becomes downward in the end, thus constraining the MFR eruption, as shown in Fig. 7c, f and Supplementary Fig. 4d. It is seen that for this event, such a non-axisymmetry induced force plays the major role in causing the failed eruption.

Another interesting finding is that the direction of the strapping force is reversed from downward at 00:16 UT to upward at 02:00 UT as shown in Supplementary Fig. 5. This means that in the later phase, the strapping force, from the external poloidal field, does not act as the main factor constraining the eruption, contrary to what is predicted in the traditional torus instability model. In fact, this active region has a quadrupole field in large scale, namely, the external poloidal field is mainly contributed by polarities P2 and N1 at 02:00 UT (long blue arrow in Supplementary Fig. 5), compared to that by P1 and N1e at 00:16 UT (long white arrow in Supplementary Fig. 5). Therefore, the external poloidal field changes its relative direction with respect to the axis of the MFR when the MFR moves upward. Consequently, the strapping force, $F_S = \mathbf{e}_z \cdot (\mathbf{J}_{TR} \times \mathbf{B}_{P,ex})$, changes its direction from downward to upward due to the change of the external poloidal field.

Previous models have mostly invoked helical kink instability and/or torus instability as the mechanisms triggering and driving the MFR eruption. The key parameters characterizing these two MHD instabilities are the twist number of the MFR and the decay index of the external poloidal field. Usually, either one of the two parameters is not sufficient for judging whether an MFR successfully erupts or not. An example is that the traditional torus instability fails to explain the final failure of some eruptions, in which the twist number decreases because of a conversion of the poloidal magnetic field to the toroidal field, but the decay index still exceeds the critical value. In such cases, the increasing tension force makes the eruptions finally failed[29]. Therefore, a

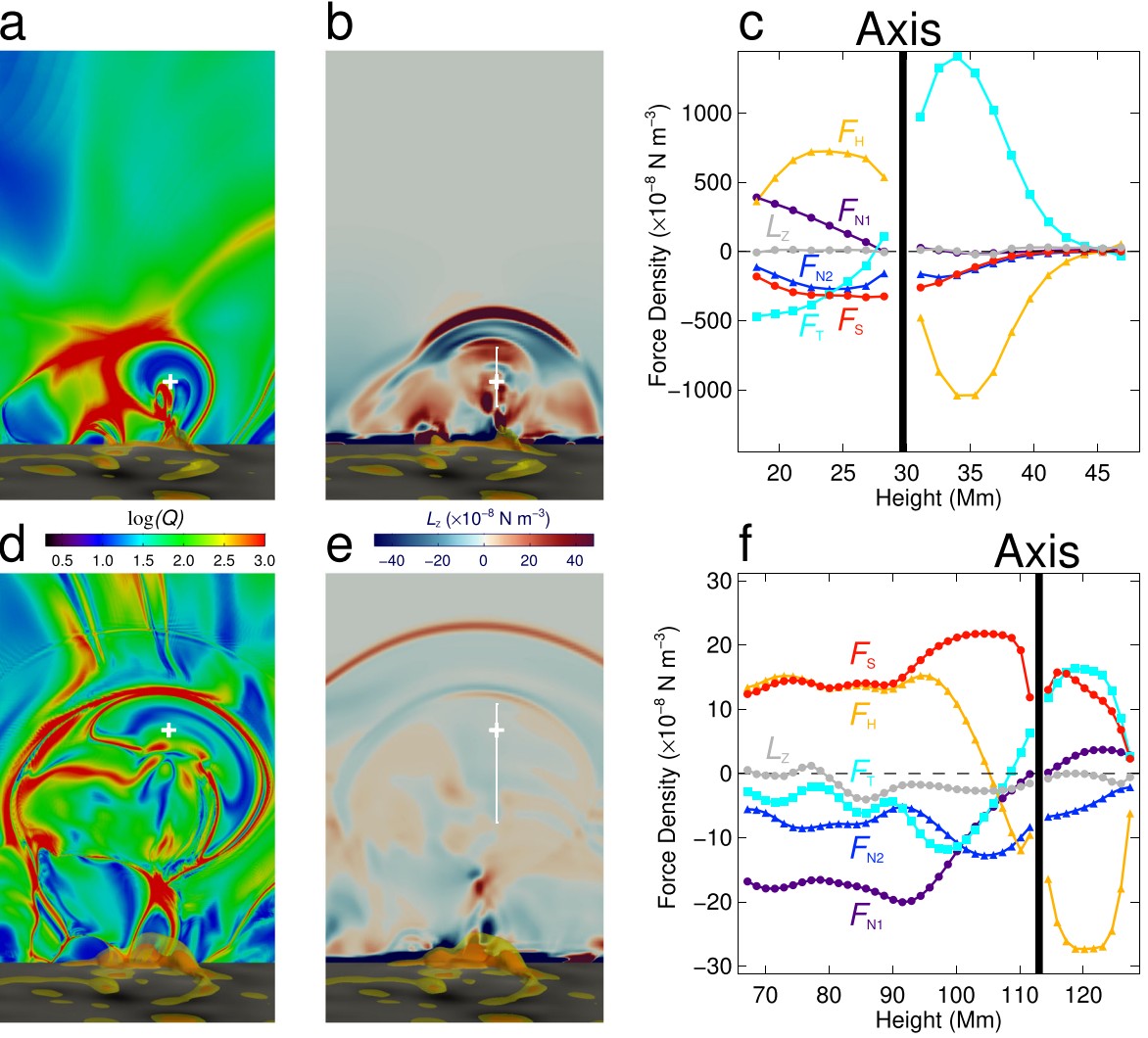

**Fig. 7 Comparison between the QSLs and the Lorentz force, $L_z$.** The upper panels are for the time of 00:16 UT and the lower panels for the time of 02:00 UT, which represent the early phase and the later phase of the eruption, respectively. **a** Distribution of the QSLs in the $Y$–$Z$ plane at the time of 00:16 UT. The QSLs show the boundary of the MFR. The white plus symbol refers to the axis of the flux rope. The background at the bottom of the figure shows the distribution of the vertical magnetic field component, $B_z$. The yellow and orange semi-transparent isosurfaces represent the electric current density larger than 21.9% and 32.9% of the maximum value in the whole domain, respectively. **b** Distribution of the Lorentz force in the $Y$–$Z$ plane at the time of 00:16 UT. The white plus symbol has the same meaning as that in **a**. The white vertical line has a length of 31.5 Mm (covering a height range from 16.8 to 48.3 Mm), which extends from the bottom to the top of the MFR. The background is the same as that in **a**. **c** Distribution of the vertical component of the Lorentz force, $L_z$, along the white line in panel **b**. The gray line is for the net force. The orange, purple, blue, red, and cyan lines represent the different components including the hoop force ($F_H$), non-axisymmetry induced forces ($F_{N1}$ and $F_{N2}$), strapping force ($F_S$) and tension force ($F_T$), which are listed in Supplementary Table 1. The black vertical line shows the location of the axis of the MFR. **d** Similar to panel **a** but at the time of 02:00 UT. **e** Similar to panel **b** but at the time of 02:00 UT. The white vertical line has a length of 63.0 Mm (covering a height range from 65.8 to 128.8 Mm). **f** Similar to panel **c** but at the time of 02:00 UT.

phase diagram has been proposed by combining the twist number and decay index[29] that can significantly improve our understanding of the eruption behaviour. Here, we also calculate the temporal evolution of the mean twist of the MFR and the decay index of the external poloidal magnetic field, which are plotted in the phase diagram of the event (Fig. 8). Based on the critical values for helical kink instability and torus instability derived from theories[18,21,22], experiments[29], observations[28,48], and MHD simulations[24,26], we divide the parameter space into four distinct regimes. We can readily find that the MFR undergoes different phases, starting from the stable regime followed by the failed kink regime, then moving to the eruptive regime and eventually stopping in the failed torus regime. In fact, it is the failed torus regime that can explain quite some failed eruptions that the

traditional torus instability cannot[48]. In conclusion, our data-driven MHD simulation provides the phase diagram for a real solar eruption and reveals the failed torus regime for confined eruptions, which has only been proposed through laboratory experiments[29]. More importantly, we propose a scenario for the failed torus regime, in which a Lorentz force component, resulting from the non-axisymmetry of the MFR, can play a major role in halting the eruption.

## Discussion

We have performed a data-driven MHD simulation using the measured photospheric magnetic field to study a realistic confined eruption. The event under investigation has a complex

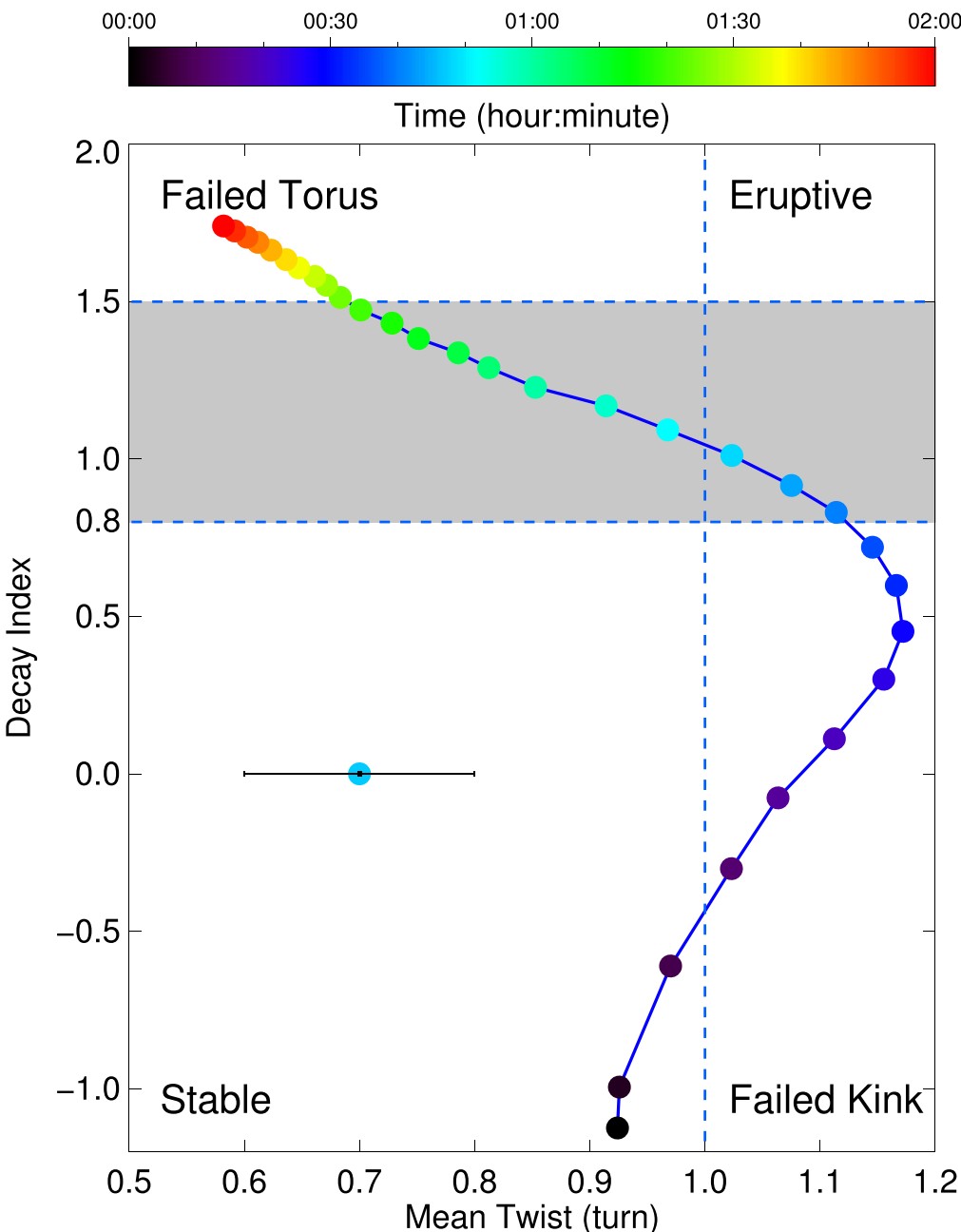

**Fig. 8 Phase diagram showing the temporal evolution of the absolute value of the mean twist of the MFR versus the decay index of the external poloidal magnetic field.** The $X$-axis represents the twist number $\mathcal{T}$ measured in turns, while the $Y$-axis represents the decay index, which are considered as basic parameters for helical kink instability and torus instability, respectively. The whole region is divided into four distinct parameter quadrants by the critical values for two instabilities. The shaded region refers to the range (the lower and upper limits) of the critical value for torus instability, as derived from theories[21–24], observations[27,28,48], or experiments[29,30]. The error bar shown in the bottom left part of the panel refers to the uncertainty of the mean twist for a typical moment.

magnetic configuration including a bifurcated MFR before the eruption onset. A good match is reached between the simulation results and observations: the magnetic field and the electric current density coincide with the extreme-ultraviolet emission both in space and time, and the dynamics of the magnetic topology is consistent with the evolving flare ribbons. Our results indicate that the MFR eruption is confined in the lower corona even if the decay index of the external poloidal field exceeds the criterion for torus instability.

Therefore, this work reports a confined eruption in the failed torus regime that is really observed and reproduced with a data-driven MHD model. Such a failed torus regime has been

proposed through laboratory experiments[29], but has rarely been found in real solar eruption events, including statistical studies[48] and case studies[27,28]. The reasons are as follows. In the ideal torus instability regime[22], the external field is assumed perpendicular to the axis of the MFR and the eruption path of the MFR is straightly upward. Thus, the horizontal component of the external magnetic field, usually from the potential field extrapolation, is often used to calculate the decay index. However, in a realistic eruption, the external field is not always perpendicular to the axis of the MFR, and the eruption path of the MFR is not always in the radial direction. Therefore, we must consider the relative orientation between the external magnetic field, the axis of the

MFR and the eruption path rather than simply adopt a horizontal field approximation[3,4,27,28,48,49]. On the other hand, previous studies are based heavily on the extrapolation model to reconstruct the coronal magnetic field for one or limited time instants, mostly before the eruption onset, which basically requires a quasi-static approximation. However, in a real situation, the magnetic field structure may undergo a significant change during the eruption process as shown in this data-driven MHD simulation. Therefore, at least for some cases, it seems invalid to use the characteristic parameters (like the twist number and decay index) at only one or few moments to judge whether an eruption can occur or not and whether it is successful or failed. Instead, a time-dependent twist number and decay index may be more appropriate in studying the dynamics of the MFR evolution. In addition, even in theory, the critical value $n_c$ of the decay index depends on the aspect ratio, as well as the ratio between the height and the footpoint distance of the MFR, which vary from case to case. Therefore, there is no universal critical value $n_c$ that can be applied to observations.

Recently, some studies suggest that the degree of electric current neutralization may be related to the ability of active regions to produce CMEs[50,51]. It provides us another view to explaining the failed eruption by observational vector magnetograms. We calculate the ratio of direct current (DC) to return current (RC) in a series of vector magnetograms for a long time containing 15 moments from 23:48 UT on 29 January 2015 to 02:36 UT on 30 January 2015. As shown in Supplementary Fig. 6a, the active region shows a coherent, middle-scale pattern of DC around its PIL. Quantitatively, the ratio |DC/RC| in both positive and negative polarities only slightly deviates from unity (Supplementary Fig. 6b). The recent statistical study[51] implies that a value of |DC/RC| that is marginally above unity as in our case cannot determine whether the active region is more likely CME-productive or CME-quiet. Therefore, prediction of flare/CME production based on the parameter |DC/RC| alone is difficult. We further measure the total axial current in the MFR as shown in Supplementary Fig. 7. One can see that the absolute values of both the DC and RC decrease rapidly while the net current increases significantly before the flare onset. After the flare peak, the RC is close to zero without obvious change while both the DC and the net current decrease gradually with time. In the end, the value of DC is reduced to only half of the original value but dominates the net current of the MFR. Our event thus displays electric current non-neutralization much more clearly in the MFR than in the whole active region.

Our simulation highlights the importance of the ideal MHD instability in driving the eruption. However, we do not preclude the role of magnetic reconnection in shaping the behaviour of the eruption. There are mainly three kinds of reconnection, namely internal reconnection between twisted field lines within the MFR body, tether-cutting reconnection between sheared field lines under the MFR, and breakout-type reconnection between the MFR and the overlying field lines. From Fig. 5b, we can judge which kinds of magnetic reconnection play the main role in different stages. First, before the flare peak, both toroidal and poloidal fluxes increase rapidly, implying a consequence of the tether-cutting reconnection. Then, the poloidal flux decreases while the toroidal flux still increases between 00:40 UT and 01:20 UT. It indicates that in this stage, although the tether-cutting reconnection still works, the internal reconnection plays a more important role, which transfers some of the poloidal flux to the toroidal flux. Finally, in the later stage, the poloidal flux keeps almost unchanged but the toroidal flux continues increasing. This implies a rough balance between the internal reconnection and the tether-cutting reconnection in changing the poloidal flux, since the former decreases but the latter increases the poloidal

flux, respectively. However, both reconnections tend to increase the toroidal flux. In all the stages, the breakout-type reconnection seems to play a minor role, since it would decrease the toroidal flux of the MFR, which is always increasing as shown in Fig. 5b. Besides, the absolute value of the mean twist number, $\mathcal{T}$, of the MFR is shown to decrease in the later two stages, which is contrary to the scenario of breakout-type reconnection that tends to increase the value of $\mathcal{T}$. In addition, the Lorentz force densities are small on the upper boundary of the MFR. This implies that even if the breakout-type reconnection would be present, it has a tiny influence on the overall distribution of the Lorentz force.

In our simulation, the field lines writhe and rotate when the MFR rises. The MFR changes its connectivity in the multipolar magnetic configuration. It initially connects polarities N1 and P1, but N1 and P2 in the end, with one footpoint changing from P1 to P2. The MFR writhing could be caused by the helical kink instability or the Lorentz force exerted on the MFR legs from the external sheared field[52]. Several plausible pieces of evidence indicate that the internal magnetic reconnection might occur in the twisted field lines if the MFR writhes and rotates. For instance, we find a 3D bald-patch separatrix surface configuration[53] in Supplementary Fig. 8a and b, showing that two typical field lines meet each other in a region depicted by a yellow dashed box. This region has a high Q value (log(Q)>3) and a high electric current density, and thus it might be a possible site for internal reconnection.

We also find a strong shear motion with a velocity up to 4 km s$^{-1}$ at the PIL. When inputting the timeseries of vector velocities to the bottom boundary to drive the MHD simulation, such a shear motion may lead to magnetic reconnection in the lower atmosphere, which produces a mildly imbalanced force and makes the MFR rise up slowly at the early stage. On the other hand, although the iteration process in the magneto-frictional method ensures that the initial condition is as close as possible to an equilibrium state, the residual force is not strictly zero, which may also incur a very slow motion of the MFR with the simulation going on.

In conclusion, our data-driven MHD model reproduces well the observations of a failed solar eruption in both the morphology of the flare and the kinematics of the MFR. The failure of the eruption can be well interpreted in terms of the failed torus regime, which cannot be explained by the traditional torus instability model. In particular, we renovate the physical framework of the failed torus regime by adding two important ingredients. We find that a force component, resulting from the radial field of the MFR, can halt the eruption, in addition to the magnetic tension force previously found[29]. The radial field is reflected by the non-axisymmetry of the MFR, which more or less exists in real events. We also find that the strapping force is not always constraining but sometimes facilitating the rise of the MFR, in particular in a quadrupole field. These findings are crucial in more properly predicting the space weather effect of CMEs that originate in complex and fast evolving magnetic structures.

## Methods

**MHD model**. Our MHD model is based on the zero-$\beta$ MHD approximation and omits the gravity and gradient of gas pressure[31], which is described as follows:

$$\frac{\partial \rho}{\partial t} + \nabla \cdot (\rho \mathbf{v}) = 0, \tag{1}$$

$$\frac{\partial (\rho \mathbf{v})}{\partial t} + \nabla \cdot (\rho \mathbf{v}\mathbf{v} + \frac{1}{2}B^2\mathbf{I} - \mathbf{B}\mathbf{B}) = 0, \tag{2}$$

$$\frac{\partial \mathbf{B}}{\partial t} + \nabla \cdot (\mathbf{v}\mathbf{B} - \mathbf{B}\mathbf{v}) = 0, \tag{3}$$

$$\mathbf{J} = \nabla \times \mathbf{B}, \tag{4}$$

where $\rho$ represents the density, $\mathbf{v}$ the velocity, $\mathbf{B}$ the magnetic field, and $\mathbf{I}$ the unit

tensor. Our model also omits density diffusion in the mass conservation equation, dynamic viscosity in the momentum conservation equation and resistivity in the magnetic induction equation. Although there is no explicit resistivity in the magnetic induction equation, magnetic reconnection still occurs due to existence of numerical resistivity. Equations (1)–(4) are written in the dimensionless form. The length, density, magnetic field, velocity and time are normalized by the characteristic quantities $L_0 = 1.0 \times 10^9$ cm, $\rho_0 = 2.3 \times 10^{-15}$ g cm$^{-3}$, $B_0 = 2.0$ G, $v_0 = B_0/(\mu_0 \rho_0)^{1/2}$ and $t_0 = L_0/v_0$, respectively. The vacuum permeability, $\mu_0$, is assumed to be unity. We solve the MHD equations numerically using the open source MPI-AMRVAC 2.0[32] and employ Powell's method[54] to clean the divergence of the magnetic field. Our computation domain has a size of $412 \times 286 \times 286$ Mm$^3$, which is divided uniformly by $288 \times 200 \times 200$ cells. The full computation box, which corresponds to $X = [-3'', 573'']$ and $Y = [-295'', 105'']$ in the $X$–$Y$ plane, is larger than the field of view shown in Fig. 2b and Supplementary Movie 1. Note that the fast mode wave produced by the eruption has not arrived at the top and side boundaries when the eruption is being confined, so that the boundary conditions can hardly affect the dynamics of the MFR.

The temperature distribution of the initial atmosphere is simplified as a stepwise function:

$$
T = \begin{cases} T_0 & 0.0 \le h < h_0 \\ k_T(h - h_0) + T_0 & h_0 \le h < h_1 \\ T_1 & h_1 \le h < 28.6 \end{cases}, \tag{5}
$$

where $T_0 = 1.4 \times 10^4$ K, $T_1 = 1.0 \times 10^6$ K, $h_0 = 3.5 \times 10^8$ cm, $h_1 = 1.0 \times 10^9$ cm, and $k_T = (T_1 - T_0)/(h_1 - h_0)$ in cgs units. Such a distribution approximately mimics the thermal structure of the actual atmosphere. The initial density profile is obtained by assuming hydrostatic equilibrium:

$$
\frac{\mathrm{d}p}{\mathrm{d}h} = -\rho g, \tag{6}
$$

where the gas pressure is expressed as $p = \rho T$ dimensionlessly. The density, $\rho$, is chosen to be $1.0 \times 10^8$ (namely, $2.3 \times 10^{-7}$ g cm$^{-3}$ before normalization) on the bottom boundary, and it decreases with height until it is fixed to be $5.0 \times 10^3$ above $h = 7.5 \times 10^9$ cm.

**Initial and boundary conditions**. The initial condition for the MHD model is provided by the nonlinear force-free field (NLFFF) derived from the magneto-frictional method[37]. On the bottom boundary, we use the vector magnetic field data at 00:00 UT (about 32 min before the onset time of the M2.0 flare) on 30 January 2015, which were observed by SDO/HMI and provided by the Joint Science Operations Center. The series name for the magnetic field data is hmi. ME_720s_fd10. Some preprocessing procedures need to be done before the magnetic field data can be used in the model, which include removal of the 180° ambiguity[55] in the transverse field, correction of the projection effect, and removal of the residual Lorentz force and torque in the original data[56]. We project the computation box to the heliocentric coordinate system in Fig. 2a and plot the reconstructed magnetic field for the initial atmosphere in Fig. 2b. The figure provides a comparison between the reconstructed field and SDO/AIA observations.

After obtaining a self-consistent initial model, we then run the MHD simulation with the bottom boundary driven by the timeseries of the vector magnetic fields and the velocity fields, which contains 11 moments from 00:00 UT to 02:00 UT with an interval of 12 min. The magnetic fields should be preprocessed as above and the velocity fields can be calculated by using the Differential Affine Velocity Estimator for Vector Magnetograms[57]. We directly set the magnetic field data to the inner ghost layer and make a zero-gradient extrapolation to the outer ghost layer of the bottom boundary. Since the vector magnetograms have been acquired with a cadence of 12 min, which is not sufficient for simulations. We then generate a new series of magnetograms with a higher cadence by linear interpolation in time. The magnetic field on the top and all side boundaries is given by the zero-gradient extrapolation. Similarly, we set the velocity data to the inner ghost layer and make a zero-gradient extrapolation to the outer ghost layer of the bottom boundary. In addition, the velocity is set to be zero on the top and four side boundaries. The density is assumed not to change on all the boundaries.

**Identification of the MFR**. An MFR consists of a bundle of helical field lines with electric currents flowing inside. First, we calculate the twist number of all sample field lines as a function of the distance, $r$, from a preassumed MFR axis. An MFR is confirmed, if there is a group of field lines with the twist number larger than 1 turn. Then, we identify the boundary of the MFR by calculating the 3D QSLs. For example, the yellow semi-transparent isosurface in Supplementary Fig. 9a delineates the 3D boundary of the MFR at 00:00 UT. We choose a high $Q$ value with $\log(Q) > 3$ as the boundary of the MFR. In a 2D view shown in Supplementary Fig. 9b, the MFR boundary is more clearly delineated by QSLs, together with some selected field lines that represent the main body of the MFR.

**Magnetic field diagnosis**. The magnetic twist[34] of an MFR is defined as

$$
\mathcal{T} = \frac{1}{2\pi} \int_s \mathbf{T}(s) \cdot \mathbf{V}(s) \times \frac{\mathrm{d}\mathbf{V}(s)}{\mathrm{d}s} \mathrm{d}s, \tag{7}
$$

where $\mathbf{T}(s)$ denotes the unit vector tangent to the axis of the MFR and $s$ is the arc length from a reference starting point on the axis curve. $\mathbf{V}(s)$ is a unit vector normal to $\mathbf{T}(s)$ and pointing from the axis curve to the secondary curve. Obviously, the twist number $\mathcal{T}$ can be computed as a function of the distance to the axis curve, which in practice can be chosen to be an arbitrary field line within the MFR. Here, we define the axis curve as the field line to which the absolute value of the average twist number is the largest. We use this largest twist number as a proxy for judging if it meets the critical condition for helical kink instability.

The decay index of the external poloidal magnetic field is defined as

$$
n = -\frac{\mathrm{d}(\log(B_{\mathrm{P,ex}}))}{\mathrm{d}(\log(r))}, \tag{8}
$$

where $B_{\mathrm{P,ex}}$ is the magnitude of the poloidal component of the external field perpendicular to the axis and the eruption path of the MFR, and $r$ is the distance measured along the eruption path of the MFR. Here, the poloidal component of the potential field at the apex of the MFR is adopted to represent the value of $B_{\mathrm{P,ex}}$ for calculating the decay index. We use this decay index as a proxy in the phase diagram to see if it reaches the threshold for torus instability.

**Reporting summary**. Further information on research design is available in the Nature Research Reporting Summary linked to this article.

## Data availability
The SDO/AIA multiwavelength data and SDO/HMI vector magnetograms can be downloaded from http://jsoc.stanford.edu/ajax/exportdata.html. The GOES X-ray flux data can be downloaded from https://www.ngdc.noaa.gov/stp/satellite/goes/dataaccess.html. All snapshots of MHD simulation data used in the present study are available on request to the corresponding author. The data for all figures are avaliable at the website https://doi.org/10.6084/m9.figshare.12949349.v2.

## Code availability
The MHD simulation code we used is the open source code of MPI-AMRVAC 2.0, which can be downloaded from its website https://github.com/amrvac/amrvac/releases/tag/v2.0/. The diagnostic tools of solar magnetic fields in the above analyses are available at the website https://github.com/njuguoyang/magnetic_modeling_codes.

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

## Acknowledgements

We thank the Solar Dynamics Observatory (SDO) team for providing the data for study. SDO is a mission of NASA's Living With a Star Program. Z.Z. acknowledges Kai Yang for his help. Z.Z., Y.G., and M.D.D. are supported by NSFC under grants 11733003, 11773016, 11533005, and 11961131002, and the China Scholarship Council 201906190107. The MHD simulations were conducted on the cluster system of the High Performance Computing Center (HPCC) in Nanjing University.

## Author contributions

Z.Z. analysed the observational data and performed model calculations. Y.G. contributed to the theoretical formulation of the model and supervised the project. M.D.D. conceived the study and supervised the project. All authors discussed the results and wrote the manuscript.

## Competing interests

The authors declare no competing interests.
