## [Peer Review File · Nature Communications]

REVIEWER COMMENTS

Reviewer #1 (Remarks to the Author):

Authors conducted a data-driven MHD simulation to reveal the dynamics of a failed torus eruption. As highlight in this study, they found new force to suppress the erupting magnetic flux rope. I highly evaluate their attempt and they quantitatively investigate the dynamics well. On the other hand, I have some concerns to their results. My comments and questions are followings.

Major comments

1. What mechanism gives first kick to the magnetic flux rope (MFR). Initial MFR is stable to kink and torus instabilities inferred from Figure 6, right? Authors used an NLFFF 44 minutes before the flare as the initial condition of the data-driven MHD simulation. Is initial condition already in non-equilibrium state? If not so, how is the magnetic field converted into the dynamics phase?
2. Interesting result that the non-axisymmetric force acting on a MFR can suppress the MFR eruption. But why does this force grow as the strong downward force during the eruption? Is this force enhanced by the nonlinear evolution of the MFR in itself? On the other hand, as far as I checked Supplementary Movie 1, the MFR looks to be destroyed through a reconnection with external field. Namely, we can consider that the hoop force is dramatically weakened by the reconnection during the eruption is essential rather than increase of the non-axisymmetric force. If the MFR is according to the latter process, this is not new finding well. In addition, it is difficult for readers, at least for me to understand the dynamics of the MFR shown in supplementary Movie1. If possible, please focus on the MFR.
3. Related to above question, authors explain the reason for decrease of the twist in Discussion "We consider that a major reason is internal magnetic reconnection...by the external reconnection". I'm not convinced this explanation. Does a reconnection easily occur inside of the MFR? For instance, the reverse field pinch (RFP) in fusion plasma research, magnetic reconnection is possible inside of the MFR because the antiparallel field lines appear there. But the physical situation is different between the erupting MFR and RFP. Please give more detailed explanation.
4. I cannot understand well why the strapping force becomes upward force. Cartoon would be helpful.

5. Regarding to Figure 6, phase diagram of the absolute value of the mean twist of the MFR vs. the decay index. According to Supplementary Figure 1, the MFR's velocity is decelerated at very early phase before $t=00:30$. This time is before the MFR can reach the gray scale area drawn in Figure 6. On the other hand, the poloidal flux still dominates over the toroidal flux around $t=00:40$ and the upward Lorentz force is also active from supplementary movie 3 around $t=00:40$. The MFR is already decelerated with some reasons before it can reach the region satisfying $n>0.8$, in this case, can we interpret as failed torus eruption? I feel like anything is off with this.

6. Regarding to simulation technique, the observed horizontal magnetic field at photosphere is fixed during the simulation, right? If so, the boundary cannot interact with the inner region consistently in the induction equation under the zero-beta MHD regime. How to remove this inconsistency, i.e., how to conserve the magnetic twist?

Minor comments

1. How to define the MFR in this study?
2. Authors should describe the observation time of the vector magnetic field that you used for the extrapolation in the main text. For instance, "44 minutes before the flare" is OK.
3. In Figures 2a and b, please add explanation of red and blue colors.

Reviewer #2 (Remarks to the Author):

This work presents a data-constrained magnetohydrodynamic (MHD) simulation for the 30 January 2015 solar eruption that caused a solar flare without coronal mass ejection (CME). The presented model was developed by using magnetic-field, extreme-ultraviolet (EUV) and soft X-ray observations for the corresponding period of time. Although this model employs the simplest zero-beta MHD approximation that ignores thermodynamic process in the solar plasma, its satisfactory accuracy has neatly been verified by comparing the resulting quasi-separatrix magnetic structure and concentration of the electric current with the observed location of the emission brightening in the volume and on the solar surface. This comparison has shown a good correspondence between them, both in terms of their locations and dynamics, which implies a relatively good quality of the developed model.

However, the paper shows a significant weakness by trying to explain the modeled failed eruption with the lack of symmetry of the configuration due to the presence of a so-called radial component of magnetic field. I am not convinced at all that this was really a cause for the failure of the eruption.

The paper actually provides a hint for the correct explanation of this fact in terms of the neutralized current in the modeled erupting flux rope. Below I provide some comments that are intended to help authors to improve their work in this respect.

1. It is not clear at all how the modeled non-axisymmetric field was decomposed on toroidal, poloidal and radial magnetic field components? The radial component with respect to what - the axis of the flux rope or the axis of the rotational symmetry in the corresponding idealized torus? Does the sum of these three components provide the total magnetic field? Unfortunately, panel (a) in supplementary Fig. 4 does not give an idea on how to answer these questions.

2. The declaration on page 9 that “a new force component, resulting from the radial field of the MFR, can halt the eruption”, is not sufficient. An explanation on how this physical effect does work is required.

3. In fact, the following comment on page 9 provides a hint on what halts the eruption in the modeled event: “In our simulation, the twist number is not uniformly distributed in the cross section of the MFR; it increases first and then decreases with the distance from the central axis, and almost approaches zero at the boundary of the MFR”. The latter implies that the total axial current in the modeled flux rope is close to zero. I think this is actually a key to understanding the modeled failed eruption. If the direct and return currents are nearly the same in absolute value, the resulting hoop force must be rather weak and incapable to sustain a CME.

4. If the axial current is indeed almost neutralized in the rope, the relevance of the critical decay index (and so of Fig. 6) becomes questionable. Because, to my best knowledge, the role of this index has been studied so far only for the flux ropes with an unneutralized axial current.

5. Nowadays, there appeared more and more articles on that how such properties of the current in the solar active regions may be related to their ability or inability to produce a CME (see, for example, Y. Liu et al 2017, ApJL 846 L6, and citations there). This literature can provide a guidance for the corresponding additional analysis that could be beneficial for the present work.

Response to Referees' Comments

Ze Zhong

zhongze@smail.nju.edu.cn

Yang Guo

guoyang@nju.edu.cn

Mingde Ding

dmd@nju.edu.cn

School of Astronomy and Space Science

Nanjing University, Nanjing, Jiangsu, China

Max Planck Institute for Solar System Research

Justus-von-Liebig-Weg 3, D-37077 Göttingen, Germany

September 13, 2020

Dear Referees:

Thanks for your comments on our manuscript entitled “The role of non-axisymmetry of magnetic flux rope in constraining solar eruptions”. We are very grateful to the referees for their careful reading and constructive comments on our paper. These comments are all valuable and helpful for revising and improving our paper. We have made a substantial revision in response to all the comments. We hope this revision can meet the requirements and answer the questions by the referees. Our revisions are shown in boldface in the new manuscript. Our detailed reply to the referees' comments is provided below.

Referee 1: Main comment

Authors conducted a data-driven MHD simulation to reveal the dynamics of a failed torus eruption. As highlight in this study, they found new force to suppress the erupting magnetic flux rope. I highly evaluate their attempt and they quantitatively investigate the dynamics well. On the other hand, I have some concerns to their results. My comments and questions are followings.

Reply: Thanks for your evaluation and suggestions on our manuscript.

1) What mechanism gives first kick to the magnetic flux rope (MFR). Initial MFR is stable to kink and torus instabilities inferred from Figure 6, right? Authors used an NLFFF 44 minutes before the flare as the initial condition of the data-driven MHD simulation. Is initial condition already in non-equilibrium state? If not so, how is the magnetic field converted into the dynamics phase?

Reply: Thanks for this important question. We are sorry that in Figure 6 of the old manuscript, we forgot to show the first three points in the diagram. We have modified Figure 6 in the new manuscript. In the early stage, the MFR is in a stable region. It didn't reach the threshold of kink or torus instabilities. We have revised the text in the last paragraph of page 8 in the new manuscript.

We confirm that our initial condition is approximately in an equilibrium state. The M2.0-class flare began at 00:32 UT and peaked at 00:44 on 30 January 2015. We built a nonlinear force-free field (NLFFF) using a magneto-frictional method for the time of 32 minutes before the flare onset. The NLFFF was run for a long time until the two metrics of force-freeness and divergence-freeness started to increase again. Then, we chose a minimum value of the force-free metric. Figure R1a in this response letter shows the variation of the two metrics against the number of iteration steps. Figure R1b shows a zoom-in view of the final 6300 iteration steps and the stopping point. The following panels (c)–(m) show the NLFFF given at different stopping points. We can see that the magnetic field has changed a lot through a magneto-frictional process before 50000 steps. After 50000 steps, the magnetic field has almost no change, and the MFR keeps unchanged suggesting that a stable state has been reached. Therefore, we constructed a field both in force-free and divergence-free

as close as possible to an equilibrium state. We revised the corresponding text in the first paragraph of page 4 in the new manuscript.

We think that there are two reasons causing the magnetic field to evolve into the dynamic phase. One is the photospheric shear motion. We find that there is a strong continuous shear motion at the polarity inversion line. The velocity reaches 4 km s^{-1} in the middle region. The shear motion leads to magnetic reconnection, so that the MFR rises up slowly due to the imbalanced force. On the other hand, although our iteration process ensures that the initial condition reaches an equilibrium state as closely as possible, the MFR may still have a small velocity due to the inevitable numerical dissipation. However, this velocity at the initial stage is much smaller compared to successful eruptions. Therefore, the kinetic evolution in the early stage of this event does not require an MHD instability. It is possible that a mild Lorentz force imbalance caused by reconnection or numerical dissipation could drive the initial dynamic evolution. Since the aim of our manuscript is to explain the final deceleration process, we do not discuss much the dynamics in the early stage in the previous version of the manuscript. We have clarified this issue in the second paragraph of page 11 in the new manuscript.

2) Interesting result that the non-axisymmetric force acting on a MFR can suppress the MFR eruption. But why does this force grow as the strong downward force during the eruption? Is this force enhanced by the nonlinear evolution of the MFR in itself? On the other hand, as far as I checked Supplementary Movie 1, the MFR looks to be destroyed through a reconnection with external field. Namely, we can consider that the hoop force is dramatically weakened by the reconnection during the eruption is essential rather than increase of the non-axisymmetric force. If the MFR is according to the latter process, this is not new finding well. In addition, it is difficult for readers, at least for me to understand the dynamics of the MFR shown in supplementary Movie1. If possible, please focus on the MFR.

Reply: First, as shown in Figures 5b and 5d in the manuscript, we decompose the total Lorentz force L_z into different components along a vertical line perpendicular to the MFR axis at two moments before and after the flare onset, respectively (another detailed decomposition cartoon for Lorentz forces is shown in the newly added Supplementary Figure 4a of the manuscript). We find that, in this case, non-axisymmetry induced forces 1 and 2 have changed with the evolution of the MFR. In particular, non-axisymmetry induced force 1 changes its direction, which is initially upward, pushing the MFR rising, but becomes downward constraining the MFR in the end, as shown in Supplementary Figure 4d. Such a dramatic change of the non-axisymmetry induced force is mainly due to the change of the topology of the MFR (in particular the radial component of the magnetic field) during its eruption. In some cases, variation of this force can become a possible reason for failed eruptions. We have added an explanation on this point in the last paragraph of page 7 in the new manuscript.

We think that the main body of the MFR is not destroyed during the eruption based on four quantities. The first is the electric current. We calculate the net electric current in a vertical plane perpendicular to the MFR axis. We show the temporal evolution of the

axial currents of the MFR in the newly added Supplementary Figure 7 in the manuscript. Before the flare onset, both positive and negative currents decrease rapidly while the net current increases. Then, the negative current changes little while both the positive and net currents decrease gradually after the flare peak. In the end, the positive current is reduced to only half of the original value and the negative current is close to zero. The second is the quasi-separatrix layers (QSLs). We superimpose the QSLs, whose squashing factor is larger than 3, on the current density distribution in Figure R2 in this response letter. The figure shows a clear boundary both in the electric current and QSLs. It illustrates that the main body of the MFR is still existing when it rises. The third is the twist number. As shown in Figure R3 in this response letter, we select four moments and plot the distribution of the twist number of some sample magnetic field lines along the distance, r , from the MFR axis. The twist number almost approaches zero at the boundary of the MFR. If magnetic reconnection occurs between the external field and the MFR at the boundary layer, the mean twist number will increase (due to stripping of low twist field lines at the boundary) but not decrease as shown in the phase diagram during the rise of the MFR. Although we do not preclude magnetic reconnection between the MFR and the external field, the internal magnetic reconnection within the body of the MFR is supposed to be the main determining factor in the change of the MFR. Such internal magnetic reconnection does not destroy the MFR. The fourth evidence is that the poloidal magnetic flux in the MFR still exists as shown in Supplementary Figure 1 of the manuscript and it implies that the MFR is not destroyed. We have mentioned these four quantities in the second paragraph of page 5 in the new manuscript.

In Supplementary Movie 1, the MFR looks like to be destroyed because we only drew some selected field lines. If we had drawn more field lines, we can see that the MFR still exists. We have improved Supplementary Movie 1 in order to display the structure of the MFR more clearly.

3) Related to above question, authors explain the reason for decrease of the twist in discussion “We consider that a major reason is internal magnetic reconnection ... by the external reconnection”. I’m not convinced this explanation. Does a reconnection easily occur inside of the MFR? For instance, the reverse field pinch (RFP) in fusion plasma research, magnetic reconnection is possible inside of the MFR because the antiparallel field lines appear there. But the physical situation is different between the erupting MFR and RFP. Please give more detailed explanation.

Reply: Thanks for this interesting point. According to the distribution of the twist number in the MFR, we consider that the variation behavior of the MFR is not dominated by the external reconnection; otherwise, it is hard to explain the decrease of the twist number of the MFR during the eruption. There are many studies based on experiments, simulations and observations suggesting that magnetic reconnection could occur inside an MFR. Myers et al. (2015) found that the transition from poloidal flux to toroidal flux occurs only when the guide field is large enough to prevent the flux rope from kinking. This relaxation occurs because the plasma can evolve to a lower energy state through internal reconfiguration

rather than through external reconnection, namely, a self-organizing behavior by magnetic reconnection. The relaxation is accompanied by an increase in the toroidal flux. In our event, as shown in Supplementary Figure 1 of the manuscript, we can see that the poloidal flux almost keeps unchanged after 01:20 UT while the toroidal flux still increases obviously, which implies an ongoing internal reconnection. In numerical simulations of a 3D model, Gibson & Fan (2006) found that the eruption of BP-grazing rope allow its partial expulsion through internal reconnection. They described in detail how the internal reconnection occurs (see their Figure 3d) by stating "each field line in the BPSS except the symmetric line at its center, has a high, arched end, and a low, dipped end. The dipped end is anchored to the photosphere, while the high arched end expands upward and is squeezed toward the MFR's vertical axis as the MFR writhes and rotates. This squeezing forms a central vertical current sheet where field lines meet and reconnect. The two high, arched halves of the reconnecting field lines form a new field line that escapes. The two low, dipped halves reconnect and settle back down as part of a lower, less twisted flux rope." A similar magnetic field configuration is found in our simulations, as depicted by the red and green field lines in the newly added Supplementary Figure 8 of the manuscript, which implies that such internal reconnections are likely to occur. From ultraviolet observations, Tripathi et al. (2009) did find many of the feature predicted by the model of Gibson & Fan (2006), such as surviving filament material and flare ribbons. Cheng et al. (2018) also reported a filament splitting-induced partial eruptions involving internal reconnection. The internal reconnection is signified by brightenings in the body of one filament and between the rising and remaining parts of both filaments. In our AIA 304 Å observations, the conjugate footpoints of the MFR displayed strong brightenings after the flare peak, indicating that the internal reconnection likely occur. In addition, Wang et al. (2017) found a flux rope with a highly twisted core and a less twisted envelope, in which the twist profile is favorable for the internal kink instability. Mei et al. (2018) found that both internal and external kink instabilities compete to drive a complex evolution of a flux rope by magnetic reconnection within and around the flux rope. In our event, the distributions of the twist number at four moments as shown in Figure R3 in this response letter reveals that the MFR has a highly twisted core with over 1.5 turns where the internal kink instability is prone to occur. We have explained more about the internal reconnection in the last paragraph of page 10 in the new manuscript.

4) I cannot understand well why the strapping force becomes upward force. Cartoon would be helpful.

Reply: We have prepared a cartoon in the newly added Supplementary Figure 5 of the manuscript. This active region has a quadrupole field in the large scale, namely, the external poloidal field is mainly contributed by polarities P2 and N1 at 02:00 UT (long blue arrow), compared to that by P1 and N1e at 00:16 UT (long white arrow). Therefore, the external poloidal field changes its relative direction with respect to the axis of the MFR when the MFR moves upward. As a consequence, the strapping force, $F_S = (\mathbf{J}_{TR} \times \mathbf{B}_{P,ex})_z$, changes its direction from downward to upward due to the change of the external poloidal field. We have revised the text in the second paragraph of page 8.

5) Regarding to Figure 6, phase diagram of the absolute value of the mean twist of the MFR vs. the decay index. According to Supplementary Figure 1, the MFRs velocity is decelerated at very early phase before $t=00:30$. This time is before the MFR can reach the gray scale area drawn in Figure 6. On the other hand, the poloidal flux still dominates over the toroidal flux around $t=00:40$ and the upward Lorentz force is also active from supplementary movie3 around $t=00:40$. The MFR is already decelerated with some reasons before it can reach the region satisfying $n > 0.8$, in this case, can we interpret as failed torus eruption? I feel like anything is off with this.

Reply: As shown in Supplementary Figure 1 of the manuscript, the MFR moves with a velocity without obvious change after its initial acceleration and is decelerated gradually after the flare peak. We check the decay index and find that the MFR reaches the region satisfying $n > 0.8$ at 00:44 UT. Therefore, after 00:44 UT, the MFR totally enters the failed torus region. However, it is true that the figure shows a tiny decrease in the MFR velocity in the early stage between 00:28 UT and 00:44 UT. There are two possible reasons for this. On the one hand, the MFR is in a failed kink regime during this period. Several statistical analyses made by Jing et al. (2018) and Duan et al. (2019) revealed that torus-stable MFR fails to erupt even if it exceeds the kink threshold. Thus, the MFR in our case may be decelerated in the failed kink regime. On the other hand, the Lorentz force acts directly on the local plasma velocity, while the apparent velocity of the MFR is related not only to the plasma velocity but also to the change of magnetic structure of the MFR. During this period, the magnetic topology has changed during the evolution of the MFR. This may explain the result that the Lorentz force is still upward but the MFR has a tiny deceleration. We have modified Figure 6 and added a description of the MFR kinematics in the second paragraph of page 5, and revised the text and added a discussion on the behavior of the MFR in the first paragraph of page 7 in the new manuscript.

6) Regarding to simulation technique, the observed horizontal magnetic field at photosphere is fixed during the simulation, right? If so, the boundary cannot interact with the inner region consistently in the induction equation under the zero-beta MHD regime. How to remove this inconsistency, i.e., how to conserve the magnetic twist?

Reply: The observed horizontal magnetic field at photosphere is not fixed. We use a data-driven (or time-varying) boundary condition including vector magnetograms and velocities to drive the evolution of the coronal magnetic field. A time series of vector magnetograms, containing 11 moments from 00:00 UT to 02:00 UT with an interval of 12 min, have been interpolated to each time step to serve as the boundary condition in the simulation. We have revised the description of the vector magnetograms in the third paragraph of page 13 in the new manuscript.

Referee 1: Minor comment

1) How to define the MFR in this study?

Reply: An MFR consists of a bundle of helical field lines with electric currents flowing inside. First, we confirm that an MFR exists by calculating the twist number of all sample field lines along the distance, r , from a pre-assumed MFR axis. An MFR is confirmed, if there is a group of field lines with the twist number larger than 1 turn. Then, we identify the boundary of the MFR by calculating the 3D quasi-separatrix layers (QSLs). The yellow and orange semi-transparent isosurfaces, in the newly added Supplementary Figure 8a of the manuscript, delineate the 3D boundary of the MFR for a sample moment at 00:00 UT. We further choose a high Q value with $\log(Q) > 3$ as the boundary of the MFR. In another view, Supplementary Figure 8b clearly shows the 2D slice of the boundary and also displays some selected field lines that represent the main body of the MFR. We have added the identification of the MFR in the Method of page 14 in the new manuscript.

2) Authors should describe the observation time of the vector magnetic field that you used for the extrapolation in the main text. For instance, “44 minutes before the flare” is OK.

Reply: Thanks for your suggestions, the M2.0-class flare occurred at 00:32 UT and peaked at 00:44 on 30 January 2015. We have modified the text in the first paragraph of page 4 and the second paragraph of page 13 in the new manuscript.

3) In Figures 2a and b, please add explanation of red and blue colors.

Reply: Figures 2a and 2b in the manuscript show the running-difference between two consecutive AIA 171 Å images in order to highlight the eruption structure. The red and blue colors represent positive and negative differences in intensity, respectively. We have added this description in the caption of Figure 2 in the second paragraph of page 4 in the new manuscript.

Referee 2: Main comment

This work presents a data-constrained magnetohydrodynamic (MHD) simulation for the 30 January 2015 solar eruption that caused a solar flare without coronal mass ejection (CME). The presented model was developed by using magnetic-field, extreme-ultraviolet (EUV) and soft X-ray observations for the corresponding period of time. Although this model employs the simplest zero-beta MHD approximation that ignores thermodynamic process in the solar plasma, its satisfactory accuracy has neatly been verified by comparing the resulting quasi-separatrix magnetic structure and concentration of the electric current with the observed location of the emission brightening in the volume and on the solar surface. This comparison has shown a good correspondence between them, both in terms of their locations and dynamics, which implies a relatively good quality of the developed model.

However, the paper shows a significant weakness by trying to explain the modeled failed eruption with the lack of symmetry of the configuration due to the presence of a so-called radial component of magnetic field. I am not convinced at all that this was really a cause for the failure of the eruption. The paper actually provides a hint for the correct explanation of this fact in terms of the neutralized current in the modeled erupting flux rope. Below I provide some comments that are intended to help authors to improve their work in this respect.

Reply: Thanks for your evaluation and suggestions on our manuscript.

1) It is not clear at all how the modeled non-axisymmetric field was decomposed on toroidal, poloidal and radial magnetic field components? The radial component with respect to what, the axis of the flux rope or the axis of the rotational symmetry in the corresponding idealized torus? Does the sum of these three components provide the total magnetic field? Unfortunately, panel (a) in supplementary Fig. 4 does not give an idea on how to answer these questions.

Reply: We have displayed a schematic picture of magnetic fields, electric currents and Lorentz forces in the MFR in the newly added Supplementary Figure 4a of the manuscript. The non-axisymmetric fields are decomposed into toroidal (\$\mathbf{B}_{T.in}\$ and \$\mathbf{B}_{T.ex}\$ ), poloidal (\$\mathbf{B}_{P.in}\$ and \$\mathbf{B}_{P.ex}\$ ) and radial (\$\mathbf{B}_{R.in}\$ and \$\mathbf{B}_{R.ex}\$ ) components, and we have verified that the sum of these three components is the total magnetic field. The radial component of the magnetic field is referred to the axis of the MFR. We have added some words to clarify this issue in the last paragraph of page 7 in the new manuscript.

2) The declaration on page 9 that “a new force component, resulting from the radial field of the MFR, can halt the eruption”, is not sufficient. An explanation on how this physical effect does work is required.

Reply: We have displayed a sketch map of different components of the Lorentz force in the newly added Supplementary Figures 4c and 4d of the manuscript. In this event, the hoop and tension forces do not change their directions during the evolution of the MFR as shown in Supplementary Figure 4c. The strapping force directs downward in the initial phase, which

constrains the MFR. Later on, it changes its direction to push the MFR rising up (as shown by a cartoon in the newly added Supplementary Figure 5 of the manuscript). These three (hoop, tension, and strapping) forces have been widely discussed in the previous literatures. However, most of them only considered symmetric flux ropes without a radial component of the magnetic field. Here, we consider a more general case with a non-axisymmetric flux rope that includes a radial component (relative to the MFR axis) of the magnetic field. This radial component includes Lorentz force components, which we call non-axisymmetry induced forces 1 and 2 (see Supplementary Table 1). We find that, the non-axisymmetry induced force 1 changes its direction during the eruption, which is initially upward, pushing the MFR rising, but becomes downward constraining the MFR in the end. Therefore, in this event, we consider that the non-axisymmetry induced force acts as a new force component to halt the eruption, which has not been mentioned in previous studies. We think that at least for some cases, this force component can be a possible reason for failed eruptions. We have described more about this force component in the last paragraph of page 7 in the new manuscript.

3) In fact, the following comment on page 9 provides a hint on what halts the eruption in the modeled event: In our simulation, the twist number is not uniformly distributed in the cross section of the MFR; it increases first and then decreases with the distance from the central axis, and almost approaches zero at the boundary of the MFR. The latter implies that the total axial current in the modeled flux rope is close to zero. I think this is actually a key to understanding the modeled failed eruption. If the direct and return currents are nearly the same in absolute value, the resulting hoop force must be rather weak and incapable to sustain a CME.

Reply: Thanks for this interesting point. We select four moments and plot the distribution of the twist number of some sample magnetic field lines along the distance, r , from the MFR axis in Figure R3 in this response letter. It is indeed that the twist number almost approaches zero at the boundary of the MFR. We then measure the total axial current in the MFR as shown in the newly added Supplementary Figure 7 of the manuscript. The purple line indicates that the axial net electric current always exists during the evolution of the MFR. The MFR is dominated by the direct current while the return current is close to zero after the flare peak. We have added a description of the temporal evolution of the axial electric current of the MFR in the first paragraph of page 10 in the new manuscript.

In the Titov–Demoulin MFR model (Titov & Démoulin, 1999), there is a clear direct current while the twist number at the boundary is close to zero (as shown in Figure 2b of Guo et al. 2017). The Titov–Demoulin MFR model has a strong non-neutralized total current because there is no return current in the model setup. This raises an interesting question why zero twist on an MFR boundary does not necessarily mean a neutralized electric current. We consider that if the system has only one current system, namely a central current, the zero twist number does imply a neutralized total current. However, in our case and in the Titov–Demoulin MFR model, the MFR has not only a strong central direct current, but also complex electric current systems below the photosphere that can balance the central current

in the MFR. Therefore, in such cases, it is possible that the twist number is close to zero on the boundary but the total axial current is not neutralized in the MFR.

4) If the axial current is indeed almost neutralized in the rope, the relevance of the critical decay index (and so of Fig. 6) becomes questionable. Because, to my best knowledge, the role of this index has been studied so far only for the flux ropes with an unneutralized axial current.

Reply: Yes, if the axial current is indeed neutralized in the MFR, the torus instability model is no longer suitable for our study. As we have mentioned above, the axial current is not neutralized. The net axial current is not zero as shown in the newly added Supplementary Figure 7 of the manuscript. We have added the description of the temporal evolution of the axial electric current of the MFR in the first paragraph of page 10 in the new manuscript.

5) Nowadays, there appeared more and more articles on that how such properties of the current in the solar active regions may be related to their ability or inability to produce a CME (see, for example, Y. Liu et al 2017, ApJL 846 L6, and citations there). This literature can provide a guidance for the corresponding additional analysis that could be beneficial for the present work.

Reply: Thanks for the valuable suggestion. We have investigated the electric current neutralization in this active region following the method proposed by Liu et al. (2017) and compared with the statistical study by Avallone & Sun (2020). In the newly added Supplementary Figure 6a of the manuscript, we display the vertical electric current density J_z at 23:48 UT on 29 January 2015, which is derived from SDO/HMI vector magnetograms. We calculate the ratio of direct current (DC) to return current (RC) in a series of vector magnetograms for a long time containing 15 moments from 23:48 UT on 29 January 2015 to 02:36 UT on 30 January 2015. In supplementary Figure 6b, we show the results of $|DC/RC|$ in both positive and negative polarities and the evolution of the unsigned magnetic flux. We can see that the DC and RC are not neutralized, though only marginally, throughout the whole time period. There is no clear tendency that the non-neutralized current evolves towards neutralization. In the statistical study by Avallone & Sun (2020), a value of $|DC/RC|$ that slightly deviates from 1 like in our case cannot clearly distinguish whether the active region is CME-productive or CME-quiet. Thus, as the authors pointed out, prediction of flare/CME production based on the parameter $|DC/RC|$ alone is difficult. We have discussed whether the degree of electric current neutralization could judge the ability to produce CMEs or not in our case in the last paragraph of page 9 in the new manuscript.

References

- Myers, C. E., Yamada, M., Ji, H., Yoo, J., Fox, W., Jara-Almonte, J., Savcheva, A., & Deluca, E. E. 2015, *Nature*, 528, 526
- Gibson, S. E., & Fan, Y. 2006, *The Astrophysical Journal*, 637, L65
- Tripathi, D., Gibson, S. E., Qiu, J., Fletcher, L., Liu, R., Gilbert, H., & Mason, H. E. 2009, *Astron. Astrophys.*, 498, 295
- Cheng, X., Kliem, B., & Ding, M. D. 2018, *The Astrophysical Journal*, 856, 48
- Wang, W., Liu, R., Wang, Y., Hu, Q., Shen, C., Jiang, C., & Zhu, C. 2017, *Nature Communications*, 8
- Mei, Z. X., Keppens, R., Roussev, I. I., & Lin, J. 2018, *A&A*, 609, A2
- Jing, J., Liu, C., Lee, J., Ji, H., Liu, N., Xu, Y., & Wang, H. 2018, *The Astrophysical Journal*, 864, 138
- Duan, A., Jiang, C., He, W., Feng, X., Zou, P., & Cui, J. 2019, *The Astrophysical Journal*, 884, 73
- Titov, V. S., & Démoulin, P. 1999, *Astron. Astrophys.*, 351, 707
- Guo, Y., et al. 2017, *The Astrophysical Journal*, 840, 40
- Liu, Y., Sun, X., Trk, T., Titov, V. S., & Leake, J. E. 2017, *The Astrophysical Journal*, 846, L6
- Avallone, E. A., & Sun, X. 2020, *The Astrophysical Journal*, 893, 123

Figure R1: **(a)** The variation of force-freeness and divergence-freeness metrics. **(b)** Zoomed-in view of two metrics for the last 6300 steps. The dotted lines represent the value of the two metrics every 1000 steps. Black arrow points the stopping point. Panels **(c)–(l)** show the nonlinear force-free field at 1000, 10000, 20000, 30000, 40000, 50000, 52000, 54000, 55000, 56000 steps, respectively. The background displays the *SDO/HMI* B_z component. The white isosurface in the bottom represents the electric current density. Cyan lines represent the dome-like structure, while green, yellow and orange lines show the main body of the MFR.

Figure R2: **Snapshots showing the axial electric current density and QSLs.** The four times 00:00, 00:44, 01:20 and 01:59 UT on 30 January 2015 correspond to the pre-flare, flare peak, post-flare and the end time of the simulation, respectively. The background image in magenta and blue displays the electric current magnitude, superimposed by a high Q value with $\log(Q) > 3$. The yellow and orange semi-transparent isosurfaces in the bottom represent the electric current density larger than 21.9% and 32.9% of the maximum value in the whole domain, respectively.

Figure R3: Distribution of the twist number of some sample magnetic field lines along the distance, r , from the MFR axis at four moments, 00:00, 00:44, 01:20 and 01:59 UT on 30 January 2015, corresponding to four moments in Figure R2, respectively.

REVIEWER COMMENTS

Reviewer #1 (Remarks to the Author):

First of all, I thank to authors for detailed explanations. Some of them convinced me. On the other hand, I am not yet fully convinced some points. My questions and comments are shown below.

Reply 1:

1. New sentence in p11 "Numerical dissipation...going on", I cannot understand this sentence. Numerical diffusion can enhance the reconnection or relax toward the lower energy state. If authors did not assume the former, dissipation would prevent the rising of the MFR and rather relax the field lines of the MFR. I think that the residual force or difference of the boundary condition between the magneto-frictional and MHD simulation is rather important. If the boundary condition is different between the magneto-frictional and MHD simulation, a physical situation of the magnetic field is also different, which might drive the MFR.

Reply2:

1. Thank you for detailed explanations and I understood that MFR is not destroyed. In previous report, "Destroy" is too strong word, so I should write "weak", sorry. Related to this topic, another question came to mind when I watched updated movie showing the dynamics of the field lines (Supplementary Movie 1). As far as I watch this movie, first the twisted field lines, which are already merged through the reconnection between the field lines shown in blue and yellow in Supplementary Figure 5, go up to the upper corona. After that, the overlying field lines in yellow in the updated movie (correspond to blue arrow in Supplementary Figure5?) are gradually twisted. This seems that the part of twist moves to the overlying field lines via reconnection. Does this depend on how to plot the field lines? However, as far as I saw the footpoints of the field lines, the twist appears as time goes where the overlying field lines were fixed before the flare. Since the erupting MFR can rotate during the eruption with the various reasons (e.g., discussed in Kliem, Török, Thompson. 2012, sol.phys. 281, 137), a reconnection is possible between the MFR and the overlying field lines. If the MFR interacts with the overlying field lines through the reconnection and releases the twist, the hoop force would be also weakened and the ascension of the MFR would stop. If the MFR is according to this scenario, I think that the halt of the rising MFR is not due to the failed torus instability.

To reply3:

1. Authors wrote that “we can see that the poloidal flux ...ongoing internal reconnection”. I think that the total flux accumulated in the MFR should be conserved even if the internal reconnection is occurred. However, in Supplemental Figure 1, the poloidal flux saturates at late phase while the toroidal flux continuously increases over time. From the view point of the flux conservation, how do we interpret this result?

2. New sentence in p 10-11: “In fact , in a...prone to occur.”

“internal kink instability is prone to occur.” Authors pointed out that the MFR is not composed of uniform twisted field lines. In this case, highly twisted lines are required to induce the kink instability (Kliem et al. 2012 sol.phys., 281, 137), probably, those would be stabilized by the less twisted field lines. In this situation, we cannot conclude that $T_w = 1.5-2.0$ is enough or not. Authors also wrote “the highly arched field...writhes and rotates”. Does this really occur? Please make sure it is done properly if authors want to explain flux transfer by the internal reconnection.

To Reply 4

Supplementary Figure 4 and Figure 5 are nice. I understood, thanks.

To Reply 5

1. The intent of my previous question is that how much a diagram shown in Figure 6 is meaningful. Because the magnetic field at around $t=00:32$ UT shown in Figure 3 is strongly deviated from the initial state while the decay index is measured from the potential field which is good approximation of the overlying field lines at initial state. Namely, the decay index value loses the meaning at $t=00:32$ UT (for example, Inoue et al. 2018 Nat. Comm. 9, 174 discussed similar things.). I think that authors need some discussions or mentions.

Reviewer #2 (Remarks to the Author):

The paper has been significantly improved by the revision; in particular, all the questions I raised have been satisfactorily resolved, so I recommend it for publication with only minor corrections suggested below.

Figure 1:

Make “white dotted box” visible in panel (b).

Caption to Supplementary Figure 4:

Change second "(c)" for "(d)".

Supplementary Table 1:

I presume that the subscript "ex" in all field components simply refers to the respective components of the potential field produced by the corresponding photospheric sources, while the subscript "in" refers to the parts of the MHD field components that are obtained after subtraction from them the "ex" components. I recommend to add a comment on this in the footnote to the table or somewhere in the main body of the article.

Response to Referees' Comments

Ze Zhong

zhongze@smail.nju.edu.cn

Yang Guo

guoyang@nju.edu.cn

Mingde Ding

dmd@nju.edu.cn

School of Astronomy and Space Science

Nanjing University, Nanjing, Jiangsu, China

Max Planck Institute for Solar System Research

Justus-von-Liebig-Weg 3, D-37077 Göttingen, Germany

November 18, 2020

Dear Referees:

We are grateful to the referees for their careful reading and constructive comments that improved our manuscript entitled “The role of non-axisymmetry of magnetic flux rope in constraining solar eruptions”. We have investigated all the points of these comments, and have revised the manuscript accordingly. All changes are shown in boldface in the new manuscript. We hope this revision can meet all requirements by the referees. Our reply to each comment raised by the referees is provided below.

Referee 1: Remarks to the Author

First of all, I thank to authors for detailed explanations. Some of them convinced me. On the other hand, I am not yet fully convinced some points. My questions and comments are shown below.

Reply: Thanks for your evaluation and helpful suggestions on our manuscript.

Reply 1:

1) New sentence in p11 “Numerical dissipation ... going on”, I cannot understand this sentence. Numerical diffusion can enhance the reconnection or relax toward the lower energy state. If authors did not assume the former, dissipation would prevent the rising of the MFR and rather relax the field lines of the MFR. I think that the residual force or difference of the boundary condition between the magneto-frictional and MHD simulation is rather important. If the boundary condition is different between the magneto-frictional and MHD simulation, a physical situation of the magnetic field is also different, which might drive the MFR.

Reply: Thanks for pointing out this important issue. We agree that the numerical dissipation has complicated effects on the MFR, including making the system evolve to a lower energy state, or affecting the dynamics of the MFR, or both. To figure out the detailed effects is difficult since we do not know where the numerical dissipation is most significant. Our expression was not clear in the old manuscript. We acknowledge that, although our iteration process ensures that the initial condition is as close as possible to an equilibrium state, the initial magnetic field still has a small but finite force-freeness metric, namely, the residual force is not strictly zero. On the other hand, we also admit that our boundary condition in the magneto-frictional (MF) process and that in the MHD simulation is different. In the MF process, we only use one preprocessed vector magnetogram at 00:00 UT as the bottom boundary rather than a time series of the preprocessed vector magnetograms from 00:00 UT to 02:00 UT with an interval of 12 min, which have been interpolated to each time step and used in the MHD simulation. Therefore, the time-varying bottom boundary can lead to a shear motion near the PIL and enhance the magnetic reconnection close to the PIL. Taking into account the above facts, we have rewritten the text in the third paragraph of page 11 in the new manuscript.

Reply 2:

1) Thank you for detailed explanations and I understood that MFR is not destroyed. In previous report, “Destroy” is too strong word, so I should write “weak”, sorry. Related to this topic, another question came to mind when I watched updated movie showing the dynamics of the field lines (Supplementary Movie 1). As far as I watch this movie, first the twisted field lines, which are already merged through the reconnection between the field lines shown in blue and yellow in Supplementary Figure 5, go up to the upper corona. After that, the overlying field lines in yellow in the updated movie (correspond to blue arrow in Supplementary Figure 5?) are gradually twisted. This seems that the part of twist moves to the overlying field lines via reconnection. Does this depend on how to plot the field lines? However, as far as I saw the footpoints of the field lines, the twist appears as time goes where the overlying field lines were fixed before the flare. Since the erupting MFR can rotate during the eruption with the various reasons (e.g., discussed in Kliem, Török, Thompson. 2012, sol.phys. 281, 137), a reconnection is possible between the MFR and the overlying field lines. If the MFR interacts with the overlying field lines through the reconnection and releases the twist, the hoop force would be also weakened and the ascension of the MFR would stop. If the MFR is according to this scenario, I think that the halt of the rising MFR is not due to the failed torus instability.

Reply: Thanks for pointing out this interesting question. We are sorry that the Supplementary Movies 1 in the old versions cannot reflect the dynamics behavior of the MFR. It is true that the old movies give an illusion that the MFR disappears or the twist is transferred from the MFR to the overlying field as time goes on. The reason is that we did not draw enough field lines on the footpoints of the MFR in the previous versions. We have improved Supplementary Movie 1 to highlight the dynamics behavior of the MFR. In the newly revised movie, one can see that the twisted MFR becomes bigger and wider when the MFR rises. The twist of the MFR always exists, and it is not transferred to the overlying field lines as time goes on.

This question gives us a motivation to classify the roles of magnetic reconnection at different places and times. There are mainly three kinds of reconnection, internal reconnection within the MFR body, external reconnection under the MFR (tether-cutting reconnection or the reconnection in a CSHKP current sheet), and external reconnection between the MFR and the overlying field lines (breakout-type). We think that the breakout-type reconnection plays a minor role here for the following reasons. According to the distribution of the twist number in Figure R1 in this response letter, the twist number almost approaches zero at the boundary of the MFR. If the breakout-type reconnection occurs in the later stage, the absolute value of the mean twist number should increase but it actually decreases as shown in the phase diagram (Figure 6). Moreover, the toroidal flux should decrease but it actually increases as shown in Supplementary Figure 1. On the other hand, as shown in the new version of Figures 5b and 5d of the new manuscript, the Lorentz force density is rather small on the upper boundary of the MFR. Therefore, even if the breakout-type reconnection were present, it has a tiny effect in weakening the hoop force. Based on these reasons, we think that the breakout-type reconnection has little effect on the MFR dynamics. We have added

a discussion on the roles of different magnetic reconnection in the last paragraph of page 10 in the new manuscript.

Reply 3:

1) Authors wrote that “we can see that the poloidal flux ... ongoing internal reconnection”. I think that the total flux accumulated in the MFR should be conserved even if the internal reconnection is occurred. However, in Supplemental Figure 1, the poloidal flux saturates at late phase while the toroidal flux continuously increases over time. From the view point of the flux conservation, how do we interpret this result?

Reply: Unlike the breakout-type reconnection, the tether-cutting reconnection does play a major role in our event, together with the internal reconnection. In our case, the magnetic flux of the MFR undergoes three stages. In the early stage, both toroidal and poloidal fluxes increase rapidly, implying a consequence of the tether-cutting reconnection. Then, the poloidal flux decreases while the toroidal flux still increases in the middle stage. It indicates that the internal reconnection plays a more important role, which transfers some of the poloidal flux to the toroidal flux (note that the total flux, poloidal flux plus toroidal flux, should indeed be conserved if merely the internal reconnection works). At last, the poloidal flux keeps almost unchanged but the toroidal flux continues increasing. This implies a rough balance between the internal reconnection and the tether-cutting reconnection in changing the poloidal flux, since the former decreases but the latter increases the poloidal flux, respectively. We have added a detailed description for the evolution of the magnetic flux and the main causes in the last paragraph of page 10 in the new manuscript.

2) New sentence in p 10-11: “In fact , in a ... prone to occur”. “internal kink instability is prone to occur”. Authors pointed out that the MFR is not composed of uniform twisted field lines. In this case, highly twisted lines are required to induce the kink instability (Kliem et al. 2012 sol.phys., 281, 137), probably, those would be stabilized by the less twisted field lines. In this situation, we cannot conclude that $Tw = 1.5-2.0$ is enough or not. Authors also wrote “the highly arched field ... writhes and rotates”. Does this really occur? Please make sure it is done properly if authors want to explain flux transfer by the internal reconnection.

Reply: Thanks for this important point. We have to distinguish the internal kink instability from the helical (or, external) kink instability. The critical twist for the internal kink instability is around 1 from the available literature. For example, in the field of laboratory plasma physics, Bergerson et al. (2006) pointed out that an internal kink instability is observed to grow when the safety factor q is below 1 (equivalently, twist number above 1). They also found that in a non-uniform condition, if q is below 1 only in the central part of the plasma, an internal instability results with formation of current singularities and ultimately reconnection. According to the distribution of the twist number in Figure R1 in this response letter, the twist number in the core is larger than 1.5 turns, which exceed the threshold of the internal kink instability. On the other hand, we need to consider the whole profile of the twist number for the cause of the helical kink instability of the MFR. As shown in Figure R1, the twist number almost approaches zero at the boundary of the MFR. Such

weakly twisted field lines would stabilize the MFR. Even if the absolute value of the mean twist number reaches to 1.2 at the flare peak, we cannot conclude whether the helical kink instability occurs or not.

In the old manuscript, we mentioned that “Our simulation does show a similar configuration where such internal reconnections are likely to occur”. We have given an additional evidence to support that internal magnetic reconnection may occur in the MFR. As shown in the newly added Supplementary Figure 8b, the yellow dotted box depicts a region where the green and red field lines meet each other. This region has a high Q value ($\log(Q) > 3$) and a high electric current density, and it is thus a preferable site for internal magnetic reconnection. We have revised the description and added the new evidence in the second paragraph of page 11 in the new manuscript.

Reply 4:

1) Supplementary Figure 4 and Figure 5 are nice. I understood, thanks.

Reply: Thanks.

Reply 5:

1) The intent of my previous question is that how much a diagram shown in Figure 6 is meaningful. Because the magnetic field at around $t=00:32$ UT shown in Figure 3 is strongly deviated from the initial state while the decay index is measured from the potential field which is good approximation of the overlying field lines at initial state. Namely, the decay index value loses the meaning at $t=00:32$ UT (for example, Inoue et al. 2018 Nat. Comm. 9, 174 discussed similar things.). I think that authors need some discussions or mentions.

Reply: The computation of the decay index depends on the potential magnetic field, the eruption path and the axis of the erupting MFR. The potential magnetic field does not change too much during the simulation period, since the normal component of the vector magnetic field on the photosphere does not change a lot as shown by observations. So, one snapshot at the initial time is acceptable. The important thing is to trace precisely the 3D evolution of the MFR, which determines the places (varying with the rising MFR) where the field component is extracted to compute the decay index. Our computation is similar to Inoue et al. (2018), who measured the decay index at a varying point tracing the MFR. In our practice, we measure the decay index at the point tracing the apex of the MFR. Of course, measuring the decay index at a fixed point above the MFR in the initial state may lose the meaning if the eruption path and the axis of the MFR are not considered. Inoue et al. (2018) also found that the local minimum of the decay index is different in the two measurements mentioned above, and pointed out that the former is consistent with observations. We have modified the description of the decay index in the Method of page 15 and revised the text in the last paragraph of page 9 in the new manuscript.

Referee 2: Remarks to the Author

The paper has been significantly improved by the revision; in particular, all the questions I raised have been satisfactorily resolved, so I recommend it for publication with only minor corrections suggested below.

Reply: Thanks for your evaluation and helpful suggestions.

Figure 1:

Make “white dotted box” visible in panel (b).

Reply: The “white dotted box” was not clear in the old manuscript. We have made it bolder and clearer in Figure 1b of the new manuscript.

Caption to Supplementary Figure 4:

Change second “(c)” for “(d)”.

Reply: We have modified it in the caption of Supplementary Figure 4 in the new manuscript.

Supplementary Table 1:

I presume that the subscript “ex” in all field components simply refers to the respective components of the potential field produced by the corresponding photospheric sources, while the subscript “in” refers to the parts of the MHD field components that are obtained after subtraction from them the example components. I recommend to add a comment on this in the footnote to the table or somewhere in the main body of the article.

Reply: Thanks for the good suggestion. We have added it in the footnote of Supplementary Table 1 in the new manuscript.

References

Bergerson, W. F., Forest, C. B., Fiksel, G., Hannum, D. A., Kendrick, R., Sarff, J. S., & Stambler, S. 2006, *Phys. Rev. Lett.*, 96, 015004

Inoue, S., Kusano, K., Bchner, J., & Skla, J. 2018, *Nat. Commun.*, 9, 174

Figure R1: Distribution of the twist number of some sample magnetic field lines along the distance, r , from the MFR axis at four moments, 00:00, 00:44, 01:20 and 01:59 UT on 30 January 2015, respectively.

REVIEWER COMMENTS

Reviewer #1 (Remarks to the Author):

Authors properly answer my questions raised in previous report. Especially, the newly added discussion on a role of the reconnection is very interesting and important to understand the dynamics of the magnetic flux rope (MFR). On the other hand, I still have questions based on author's response. My questions and comments are shown below.

To Reply2: The update movie is nice which represents the formation process of the coherent MFR well. But this movie brought another question to me. Figure 1c shows the vector magnetic field and the magnetic field lines. Is this full size of the numerical simulation? If so, the newly formed MFR looks to be large compared to the size of the numerical domain. I am wondering how much do the boundary condition (side and top) affect the growth of the MFR. In other words, is there possible that these boundary conditions control the upward velocity and the writhing of the MFR during the growth? Since this simulation imposes zero-gradient extrapolation and the velocity is fixed to be zero at the side and top boundaries. I think that this condition looks to behavior as a wall which strongly restricts the dynamics of the MFR.

To Reply3-1: The newly added discussion described in the last paragraph of page 10 is interesting. I am beginning to understand the growth process of the magnetic flux of the MFR. The point that I don't understand yet is what is the physical difference between the internal reconnection and the tether-cutting reconnection. From the manuscript, both seem to be reconnection between the sheared field lines but the name is different. This may confuse the readers.

To Reply3-2: The last sentence of second paragraph of p 11 (This region has a high Q value...to be induced) looks strong for me. Although authors referred Bergerson et al. 2006 and discussed based on their results, I think that the results (quantitative value) from Bergerson et al 2006 cannot be directly applicable to presented results because the physical situation is different between them. Probably, Bergerson et al. 2006 conducted the stability analysis using static magnetic field while the MFR presented by authors is already in the dynamic phase. Therefore, isn't the growth rate of the instability different between Bergerson et al. 2006 and this study. In other word, when an MFR is far from the static state, I doubt that the twist number 1-2 is really enough to cause the kink instability. Of course, the kink instability might be one of candidates, but authors should mention another candidates too. The kink instability is not only way to cause the writhing of the MFR(See Kliem et al.2012 Sol.Phys., 281, 137).

Unfortunately, it is difficult for me to recognize the Supplementary Figure 8(b) as the evidence of the internal reconnection. Since the helical twisted lines are concentrated in the yellow dotted square, is the current density intense broadly? That does not necessarily mean that the reconnection takes place. If authors check the temporal evolution of the current density, it might answer this question.

To Reply 5: Thank you for polite explanation.

Miner comments

1. Supplementary Figure 8(b): Please make the yellow dotted line thick.
2. Method: In equation(2), why is $\nabla\{B^2/2\}$ isolated ? Do you have some reason not to described perfect conservation form?
3. Method: How to remove error of Div B during the simulation? If authors treat something, it is better to explain with short sentence.

Response to Referee's Comments

Ze Zhong

zhongze@smail.nju.edu.cn

Yang Guo

guoyang@nju.edu.cn

Mingde Ding

dmd@nju.edu.cn

School of Astronomy and Space Science

Nanjing University, Nanjing, Jiangsu, China

Max Planck Institute for Solar System Research

Justus-von-Liebig-Weg 3, D-37077 Göttingen, Germany

January 24, 2021

Dear Referee:

We thank the referee for his/her constructive comments and helpful suggestions that helped to improve our manuscript entitled "The role of non-axisymmetry of magnetic flux rope in constraining solar eruptions". We agree with the referee's comments and have revised the manuscript accordingly. We hope this revision can meet the requirements and answer the questions by the referee. Our revisions in the new manuscript are shown in boldface. The detailed reply to each comment is provided below.

Referee 1: Main comment

Authors properly answer my questions raised in previous report. Especially, the newly added discussion on a role of the reconnection is very interesting and important to understand the dynamics of the magnetic flux rope (MFR). On the other hand, I still have questions based on authors response. My questions and comments are shown below.

Reply: Thanks for your evaluation and helpful suggestions on our manuscript.

To Reply2:

1) The update movie is nice which represents the formation process of the coherent MFR well. But this movie brought another question to me. Figure 1c shows the vector magnetic field and the magnetic field lines. Is this full size of the numerical simulation? If so, the newly formed MFR looks to be large compared to the size of the numerical domain. I am wondering how much do the boundary condition (side and top) affect the growth of the MFR. In other words, is there possible that these boundary conditions control the upward velocity and the writhing of the MFR during the growth? Since this simulation imposes zero-gradient extrapolation and the velocity is fixed to be zero at the side and top boundaries. I think that this condition looks to behavior as a wall which strongly restricts the dynamics of the MFR.

Reply: We are sorry that we did not describe the field of view (FOV) of the movie in the previous version. To highlight the MFR, we only display a central part of the full computation box in Supplementary Movie 1 and Figure 1c of the manuscript. Our full computation domain is much larger. Figure R1 in this response letter shows the full computation domain at four selected snapshots. Our computation box has a size of \$412 \times 286 \times 286\$ Mm³, corresponding to \$X = [-3'', 573'']\$ and \$Y = [-295'', 105'']\$ in the X-Y plane, while the FOV of Figure 1c is \$X = [130'', 440'']\$ and \$Y = [-210'', 10'']\$. We have mentioned this difference in the Method section of page 12 and the caption of Supplementary Movie 1 in the new manuscript.

Moreover, in the Z-direction, the final height of the MFR is less than half the height of the computation box. That is to say, the size of the MFR is much smaller than the size of the computation box in all the three dimensions, so that the boundary conditions can hardly affect the dynamics of the MFR. We have further checked and confirmed that the boundary conditions have little effect on the upward motion and writhing of the MFR.

To Reply3–1:

1) The newly added discussion described in the last paragraph of page 10 is interesting. I am beginning to understand the growth process of the magnetic flux of the MFR. The point that I don't understand yet is what is the physical difference between the internal reconnection and the tether-cutting reconnection. From the manuscript, both seem to be reconnection between the sheared field lines but the name is different. This may confuse the readers.

Reply: Thanks for pointing out this issue. We tend to explain more clearly the difference between the internal reconnection and the tether-cutting reconnection. The internal reconnection occurs between two twisted field lines which belong to a part of the MFR. The footpoints of two reconnecting field lines are located at the footpoint regions of the MFR. However, the tether-cutting reconnection occurs between two sheared field lines under the MFR body. In the latter case, the two reconnecting field lines do not belong to the MFR before the reconnection. After the reconnection, one reconnected field line becomes a part of the MFR, while the other is far away from the MFR. We have clarified the confusing wording in the previous version and defined more clearly the different kinds of reconnection in the last paragraph of page 10 in the new manuscript.

To Reply3–2:

1) The last sentence of second paragraph of p11 (This region has a high Q value ... to be induced) looks strong for me. Although authors referred Bergerson et al. (2006) and discussed based on their results, I think that the results (quantitative value) from Bergerson et al. (2006) cannot be directly applicable to presented results because the physical situation is different between them. Probably, Bergerson et al. (2006) conducted the stability analysis using static magnetic field while the MFR presented by authors is already in the dynamic phase. Therefore, isn't the growth rate of the instability different between Bergerson et al. (2006) and this study. In other word, when an MFR is far from the static state, I doubt that the twist number 1–2 is really enough to cause the kink instability. Of course, the kink instability might be one of candidates, but authors should mention another candidates too. The kink instability is not only way to cause the writhing of the MFR (See Kliem et al. 2012 Sol. Phys., 281, 137).

Reply: Thanks for this important point. We agree that our proposed evidence supporting the occurrence of the internal reconnection is only qualitatively possible, but not fully justified. As pointed out by the referee, although the internal kink instability is observed to grow when the safety factor q is below 1 in the static state, it is difficult to confirm that the kink instability really occurs in the dynamic phase with a twist number 1–2. Therefore, we have softened the statement on the evidence of the internal reconnection. We also agree that the kink instability is only one possible candidate to cause the writhing of the MFR. We have checked the external sheared field component, \mathbf{B}_{ex} , and found that the Lorentz force exerted by the sheared field component proposed by Kliem et al. (2012) also plays a role in writhing the MFR, thus confirming the referee's point. We have added the role of

the sheared field in the 2nd paragraph of page 11 in the new manuscript.

2) Unfortunately, it is difficult for me to recognize the Supplementary Figure 8(b) as the evidence of the internal reconnection. Since the helical twisted lines are concentrated in the yellow dotted square, is the current density intense broadly? That does not necessarily mean that the reconnection takes place. If authors check the temporal evolution of the current density, it might answer this question.

Reply: We agree that it is difficult to find the precise location of the internal reconnection. There is still no much knowledge on the internal reconnection. It may occur in a relatively broad region (with a relatively large current density) within the MFR, but not presenting a long thin current sheet as in the standard flare/CME model.

We have checked the temporal evolution of the current density for the region with a high current density value as shown in Figure R2 in this response letter. Only the stage after the flare peak is analyzed, when the internal reconnection may play an important role. We find that both the average and maximum current densities show a coherent decrease. It implies that the location of the high current density might be a possible site for the internal reconnection since the electric current is gradually dissipated. However, this is only a reasonable speculation. Therefore, We have weakened our statement on the internal magnetic reconnection in the 2nd paragraph of page 11 in the new manuscript.

Referee 1: Minor comment

1) Supplementary Figure 8(b): Please make the yellow dotted line thick.

Reply: We have made the yellow dotted line bolder and clearer in Supplementary Figure 8b of the new manuscript.

2) Method: In equation(2), why is $\nabla(B^2/2)$ isolated? Do you have some reason not to described perfect conservation form?

Reply: Thanks for your suggestions. The term $\nabla(B^2/2)$ can be written as a part of the conservation form. It is equivalent to $\nabla \cdot (\mathbf{I}B^2/2)$. We have modified equation (2) in the Method section of page 12 in the new manuscript.

3) Method: How to remove error of Div B during the simulation? If authors treat something, it is better to explain with short sentence.

Reply: In the MHD simulation, we use Powell's method (Powell et al., 1999) to clean the divergence of the magnetic field. We have mentioned it in the Method section of page 12 in the new manuscript.

References

- Bergerson, W. F., Forest, C. B., Fiksel, G., Hannum, D. A., Kendrick, R., Sarff, J. S., & Stambler, S. 2006, Phys. Rev. Lett., 96, 015004
- Kliem, B., Török, T., & Thompson, W. T. 2012, Sol. Phys., 281, 137
- Powell, K. G., Roe, P. L., Linde, T. J., Gombosi, T. I., & De Zeeuw, D. L. 1999, J. Comput. Phys., 154, 284

Figure R1: **Snapshots showing the MFR and the electric current density.** The four times 00:00, 00:32, 00:44 and 01:20 UT on 30 January 2015 correspond to the pre-flare, flare onset, flare peak and end times, respectively. The vertical transparent slice displays the electric current density. The cyan, yellow and orange lines have the same meaning as that in Figure 1c of the manuscript, only that more field lines are drawn here. The olive lines refer to field lines connecting the negative polarity N1 and the positive polarity P2 at the first moment. The white isosurface represents the electric current density larger than 32.9% of the maximum value in the whole domain. The background shows the distribution of the vertical magnetic field component, B_z . The computation box has a size of $412 \times 286 \times 286 \text{ Mm}^3$, corresponding to $X = [-3'', 573'']$ and $Y = [-295'', 105'']$ in the X-Y plane.

Figure R2: **Temporal evolution of the electric current density for the region with a high current density value (the yellow dashed box of Supplementary Figure 8) after the flare peak.** The red and orange lines represent the average and maximum electric current densities, respectively. The green vertical line shows the peak time of the M2.0 flare at 00:44 UT.

REVIEWER COMMENTS

Reviewer #1 (Remarks to the Author):

I first thank authors for making effort to revise the draft and addressing appropriate reply. I understand much better now, so I am ready to recommend the publication from the Nature Communication. However, I am sorry but I have a few more questions to your reply.

To Reply2: "Moreover, in the Z-direction, the final height of the MFR is less than half the height of the computation box. That is to say, the size of the MFR is much smaller than the size of the computation box in all the three dimensions, so that the boundary conditions can hardly affect the dynamics of the MFR. We have further checked and confirmed that the boundary conditions have little effect on the upward motion and writhing of the MFR."

I feel that this explanation might be a little bit rough. Why does the final height of the MFR is less than half the height of the computation box is enough not to be affected by the boundary condition? What is the evidence? The arc shape structure above the MFR corresponds to the fast mode wave? If the fast mode wave cannot touch the boundary, i.e., the physical information does not across the whole numerical box yet, authors might state that the top and side boundaries do not work on the dynamics of the MFR. What do you think?

To Reply3-2: We have checked the temporal evolution of the current density for the region with a high current density value as shown in Figure R2 in this response letter. Only the stage after the flare peak is analyzed, when the internal reconnection may play an important role. We find that both the average and maximum current densities show a coherent decrease. It implies that the location of the high current density might be a possible site for the internal reconnection since the electric current is gradually dissipated. However, this is only a reasonable speculation. Therefore, We have weakened our statement on the internal magnetic reconnection in the 2nd paragraph of page 11 in the new manuscript.

Thank you very much for your reply and making Figure R2.

Response to Referee's Comments

Ze Zhong

zhongze@smail.nju.edu.cn

Yang Guo

guoyang@nju.edu.cn

Mingde Ding

dmd@nju.edu.cn

*School of Astronomy and Space Science
Nanjing University, Nanjing, Jiangsu, China*

March 04, 2021

Dear Referee:

We thank the referee very much for his/her careful reading and constructive comments that helped to improve our manuscript entitled "The role of non-axisymmetry of magnetic flux rope in constraining solar eruptions". We agree with the referee's comments and have made a revision of the manuscript accordingly. We hope this revision can meet the requirements and answer the questions by the referee. Our revisions in the new manuscript are shown in boldface.

Referee 1: Comment

I first thank authors for making effort to revise the draft and addressing appropriate reply. I understand much better now, so I am ready to recommend the publication from the Nature Communication. However, I am sorry but I have a few more questions to your reply.

Reply: Thanks for your evaluation and helpful suggestions on our manuscript.

To Reply2:

1) "Moreover, in the Z-direction, the final height of the MFR is less than half the height of the computation box. That is to say, the size of the MFR is much smaller than the size of the computation box in all the three dimensions, so that the boundary conditions can hardly affect the dynamics of the MFR. We have further checked and confirmed that the boundary conditions have little effect on the upward motion and writhing of the MFR."

I feel that this explanation might be a little bit rough. Why does the final height of the MFR is less than half the height of the computation box is enough not to be affected by the boundary condition? What is the evidence? The arc shape structure above the MFR corresponds to the fast mode wave? If the fast mode wave cannot touch the boundary, i.e., the physical information does not across the whole numerical box yet, authors might state that the top and side boundaries do not work on the dynamics of the MFR. What do you think?

Reply: Thanks for the comments. The arc shape structure above the MFR indeed corresponds to the fast mode wave, but with an Alfvén speed, since the zero-beta assumption omits the thermal pressure. We have calculated the speed of the wave. According to the speed distribution, we can calculate the propagation distance of the wave. We found that the wave only propagates to a height of about 162 Mm at 01:59 UT (see Figure R1a in this response letter), when the simulation ends, while the computation box has a height of 286 Mm. In the horizontal direction, we found that the wave travels to the side boundaries at 01:52 UT (see Figure R1b in this response letter), while the confinement of the eruption occurs much earlier at about 00:52 UT (see Supplementary Figure 1 in the manuscript). This indicates that the wave has not reached the top and side boundaries when the eruption is being confined. Therefore, we can conclude that the boundary conditions can hardly affect the dynamics of the MFR. We have mentioned this issue in the Method section of page 12 in the new manuscript.

To Reply3–2:

1) “We have checked the temporal evolution of the current density for the region with a high current density value as shown in Figure R2 in this response letter. Only the stage after the flare peak is analyzed, when the internal reconnection may play an important role. We find that both the average and maximum current densities show a coherent decrease. It implies that the location of the high current density might be a possible site for the internal reconnection since the electric current is gradually dissipated. However, this is only a reasonable speculation. Therefore, We have weakened our statement on the internal magnetic reconnection in the 2nd paragraph of page 11 in the new manuscript.”

Thank you very much for your reply and making Figure R2.

Reply: Thanks.

Figure R1: Distribution of the electric current density showing the wave positions (indicated by arrows) in the vertical direction at 01:59 UT and in the horizontal direction at 01:52 UT.

REVIEWERS' COMMENTS

Reviewer #1 (Remarks to the Author):

The physics of the solar eruption is one of long-standing problems in the solar physics. The torus instability is one of strong candidates to explain an onset of solar eruptions and an initiation of coronal mass ejections (CMEs). For the past 10 years, many solar physicists believe the initiation of the solar eruptions and it grows into the CME if the magnetic flux rope (MFR) satisfies a threshold of the torus instability. On the other hand, more recently, Myers et al. 2015 Nature and Inoue et al. 2018 Nat. Com. claimed that the threshold is effective for the initial kick of the MFR for the eruption but no guarantee in the nonlinear evolution, i.e., growth of the CME.

This paper numerically reproduces the failed eruption and clears the issue of why the eruption is halted even though the MFR satisfies the threshold of the torus instability. The outstanding of this paper is to find new force, which evolves in the nonlinear regime of the MFR dynamics, to suppress the MFR eruption. This result suggests that the physics of the solar eruption is never simple, i.e., it is not governed only by the threshold of the torus instability (decay index). The analysis presented in this paper is conducted very carefully and meticulously. Therefore, the presented results are reliable.

From these reason, this work is definitely important and would greatly contribute to the solar physics community. Therefore, I recommend this work publication from Nature Communications.

Finally, I sincerely thank authors to answer my questions. I am looking forward to next your work!

Response to Referee's Comments

Ze Zhong

zhongze@smail.nju.edu.cn

Yang Guo

guoyang@nju.edu.cn

Mingde Ding

dmd@nju.edu.cn

*School of Astronomy and Space Science
Nanjing University, Nanjing, Jiangsu, China*

March 31, 2021

Referee 1: Comment

The physics of the solar eruption is one of long-standing problems in the solar physics. The torus instability is one of strong candidates to explain an onset of solar eruptions and an initiation of coronal mass ejections (CMEs). For the past 10 years, many solar physicists believe the initiation of the solar eruptions and it grows into the CME if the magnetic flux rope (MFR) satisfies a threshold of the torus instability. On the other hand, more recently, Myers et al. (2015) and Inoue et al. (2018) claimed that the threshold is effective for the initial kick of the MFR for the eruption but no guarantee in the nonlinear evolution, i.e., growth of the CME.

This paper numerically reproduces the failed eruption and clears the issue of why the eruption is halted even though the MFR satisfies the threshold of the torus instability. The outstanding of this paper is to find new force, which evolves in the nonlinear regime of the MFR dynamics, to suppress the MFR eruption. This result suggests that the physics of the solar eruption is never simple, i.e., it is not governed only by the threshold of the torus instability (decay index). The analysis presented in this paper is conducted very carefully and meticulously. Therefore, the presented results are reliable.

From these reason, this work is definitely important and would greatly contribute to the solar physics community. Therefore, I recommend this work publication from Nature Communications.

Finally, I sincerely thank authors to answer my questions. I am looking forward to next your work!

Reply: Thank you very much for your favorable comments on our manuscript. We will continue the work in this field.

References

- Myers, C., Yamada, M., Ji, H., Yoo, J., Fox, W., Jara-Almonte, J., Savcheva, A., & Deluca, E. 2015, *Nature*, 528, 526
- Inoue, S., Kusano, K., Bchner, J., & Skla, J. 2018, *Nat. Commun.*, 9, 174